



# Robust observational constraint of uncertain aerosol processes and emissions in a climate model and the effect on aerosol radiative forcing

Jill S. Johnson[1], Leighton A. Regayre[1], Masaru Yoshioka[1], Kirsty J. Pringle[1], Steven T. Turnock[2], Jo Browse[3], David M. H. Sexton[2], John W Rostron[2], Nick A. J. Schutgens[4], Daniel G. Partridge[5], Dantong Liu[6,#], James D. Allan[6,7], Hugh Coe[6], Aijun Ding[8], David D. Cohen[9], Armand Atanacio[9], Ville Vakkari[10,11], Eija Asmi[10] and Ken S. Carslaw[1]

[1]Institute for Climate and Atmospheric Science, School of Earth and Environment, University of Leeds, Leeds, UK
[2]Met Office Hadley Centre, Exeter, UK
[3]Centre for Geography and Environmental Science, University of Exeter, Penryn, UK
[4]Earth Sciences, Faculty of Science, Vrije Universiteit Amsterdam, Amsterdam, Netherlands
[5]College for Engineering, Mathematics, and Physical Science, University of Exeter, Exeter, UK
[6]Centre for Atmospheric Sciences, School of Earth and Environmental Sciences, University of Manchester, Manchester, UK
[7]National Centre for Atmospheric Science, University of Manchester, Manchester, UK
[8]Joint International Research Laboratory of Atmospheric and Earth System Sciences (JirLATEST), School of Atmospheric Sciences, Nanjing University, Nanjing 210023, China
[9]Centre for Accelerator Science, ANSTO, New Illawarra Rd, Lucas Heights, NSW, 2232, Australia
[10]Finnish Meteorological Institute, Helsinki, FI-00101, Finland
[11]Unit for Environmental Sciences and Management, North-West University, Potchefstroom, ZA-2520, South Africa

[#]now at: Department of Atmospheric Sciences, School of Earth Sciences, Zhejiang University, Hangzhou, Zhejiang, China.

*Correspondence to*: Jill S. Johnson (J.S.Johnson@leeds.ac.uk)

**Abstract.** The effect of observational constraint on the ranges of uncertain physical and chemical process parameters was explored in a global aerosol-climate model. The study uses 1 million variants of the HadGEM3-UKCA climate model that sample 26 sources of uncertainty, together with over 9000 monthly aggregated grid-box measurements of aerosol optical depth, PM2.5, particle number concentrations, sulphate and organic mass concentrations. Despite many compensating effects in the model, the procedure constrains the probability distributions of parameters related to secondary organic aerosol, anthropogenic $SO_2$ emissions, residential emissions, sea spray emissions, dry deposition rates of $SO_2$ and aerosols, new particle formation, cloud droplet pH and the diameter of primary combustion particles. Observational constraint rules out nearly 98% of the model variants. On constraint, the $\pm 1\sigma$ (standard deviation) range of global annual mean direct radiative forcing, $RF_{ari}$, is reduced by 33% to -0.14 to -0.26 W m$^{-2}$, and the 95% credible interval (CI) is reduced by 34% to -0.1 to -0.32 W m$^{-2}$. For the global annual mean aerosol-cloud radiative forcing, $RF_{aci}$, the $\pm 1\sigma$ range is reduced by 7% to -1.66 to -2.48 W m$^{-2}$, and the 95% CI by 6% to -1.28 to -2.88 W m$^{-2}$. The tightness of the constraint is limited by parameter cancellation effects (model equifinality) as well as the large and poorly defined 'representativeness error' associated with comparing point measurements with a global model. The constraint could also be narrowed if model structural errors that prevent simultaneous agreement with different measurement types in multiple locations and seasons could be improved. For example, constraints using either sulphate or

PM2.5 measurements individually result in RF$_{ari}$ ±1σ ranges that only just overlap, which shows that emergent constraints based on one measurement type may be over-confident.

## 1    Introduction

Global model simulations of aerosols and their climatic effects are very uncertain. Different global aerosol models have large
spread in their simulations of aerosol microphysics, radiation and forcing (Mann et al., 2014; Myhre et al., 2013; Shindell et al., 2013; Tsigaridis et al., 2014). This multi-model spread can be due to different model structures, missing processes, parameter settings, algorithms or coding errors. Individual climate models are also very uncertain because the values of parameters related to physical processes and emissions are often poorly defined (Johnson et al., 2018; Lee et al., 2011b, 2013; Regayre et al., 2018). The uncertainty in the aerosol effective radiative forcing (ERF) over the industrial period caused by
aerosol processes, physical atmosphere model processes and emissions could be as large as the multi-model spread (Johnson et al., 2018; Regayre et al., 2014, 2018).

There are two main methods to reduce model uncertainty, often called bottom-up and top-down approaches. The bottom-up approach involves improving the representation of model processes and refining estimates of the associated parameter values
through experiment and theory. This approach is necessary to improve model fidelity, but it does not provide an estimate of the model uncertainty, and the uncertainty may grow if the increase in model complexity requires a large number of new and poorly defined parameters. To reduce model uncertainty, bottom-up model development needs to be combined with top-down approaches in which numerous uncertain process-related parameters and emissions are adjusted to improve the agreement of models with measurements.

The difficulty with top-down model adjustments (in its simplest form, model tuning) is that the uncertainty stems from large combinations of uncertain model input parameters. This means that the adjustment of small sets of parameters to improve model agreement with measurements will not produce robust results (Carslaw et al., 2018). For example, a model simulation of particle concentrations could be improved by adjusting particle formation rates, but many other combinations of parameters
related to emissions, chemistry or deposition might be able to achieve similar model skill (Carslaw et al., 2013b). Models that are narrowly tuned in this way can therefore produce a wide range of results when used to make predictions outside the range of conditions under which they were tuned. This is likely to be a cause of the large uncertainty in aerosol radiative forcing, which is a predicted rather than observable quantity.

If other aerosol-climate models are comparable with our own HadGEM-UKCA model, then they contain at least 20 important uncertain parameters related to emissions and processes, although fewer than about 10 parameters will dominate the uncertainty in a particular model variable in any one environment and time of year (Lee et al., 2016; Regayre et al., 2014,

2018). Therefore, to define and reduce the model uncertainty it is necessary to find from within 10 dimensions of parameter space all the parameter combinations that produce plausible agreement with different aerosol properties observed across all seasons and global environments. A single well-configured version of a model produced by parameter tuning tells us nothing about the combinations of parameter values that can achieve consistency with measurements within their uncertainty range, nor does it tell us anything about the model output uncertainty.

In this paper we address the question: To what extent do extensive and diverse aerosol measurements enable the plausible range of model parameters to be constrained if the full range of their compensating effects is accounted for? By 'constrain' we mean a narrowing of the probability distribution of a parameter (and potentially the absolute range) compared to the uncertainty range that was assumed when the model was built. We also quantify how the identification of observationally plausible parameter ranges feeds through to a reduction in the uncertainty in predictions of aerosol radiative forcing over the industrial period. The study focuses on model constraint using measurements of aerosol properties rather than cloud properties, therefore we emphasise the effect on aerosol-radiation interaction forcing rather than aerosol-cloud interaction.

## 2 Methods

### 2.1 The HadGEM3-UKCA climate model

We use the Global Atmosphere 4 (GA 4.0; Walters et al., 2014) configuration of the Hadley Centre General Environment Model version 3 (HadGEM3; Hewitt et al., 2011), which incorporates the UK Chemistry and Aerosol (UKCA) model at version 8.4 of the UK Met Office's Unified Model (UM). UKCA simulates trace gas chemistry and the evolution of the aerosol particle size distribution and chemical composition using the GLObal Model of Aerosol Processes (GLOMAP-mode; Mann et al., 2010) and a whole-atmosphere chemistry scheme (Morgenstern et al., 2009; O'Connor et al., 2014). The model has a horizontal resolution of 1.25x1.875 degrees and 85 vertical levels.

The aerosol size distribution is defined by seven log-normal modes: one soluble nucleation mode as well as soluble and insoluble Aitken, accumulation and coarse modes. The aerosol chemical components are sulphate, sea salt, black carbon (BC), organic carbon (OC) and dust. The model does not include any representation of nitrate aerosols. Secondary organic aerosol (SOA) material is produced from the first stage oxidation products of biogenic monoterpenes under the assumption of zero vapour pressure. SOA is combined with primary particulate organic matter after kinetic condensation.

GLOMAP simulates new particle formation, coagulation, gas-to-particle transfer, cloud processing and deposition of gases and aerosols. The activation of aerosols into cloud droplets is calculated using globally prescribed distributions of sub-grid vertical velocities (West et al., 2014) and the removal of cloud droplets by autoconversion to rain is calculated by the host model. Aerosols are also removed by impaction scavenging of falling raindrops according to the parametrisation of clouds and





precipitation collocation (Boutle et al., 2014; Lebsock et al., 2013). Aerosol water uptake efficiency is determined by kappa-Kohler theory (Petters and Kreidenweis, 2007) using composition-dependent hygroscopicity factors.

Anthropogenic emission scenarios prepared for the Atmospheric Chemistry and Climate Model Inter-comparison Project (ACCMIP) and prescribed in some of the CMIP Phase 5 experiments are used here. Biomass burning emissions for recent decades were prescribed using a ten year average of 2002 to 2011 monthly mean data from the Global Fire and Emissions Database (GFED3; van der Werf et al., 2010) and according to Lamarque et al. (2010) for 1850. Volcanic $SO_2$ emissions are prescribed in the model by combining emissions from the Andres and Kasgnoc (1998) dataset for continuously erupting and sporadically erupting volcanoes and the Halmer et al. (2002) dataset for explosive volcanoes.

A full description of the set-up for our model simulations can be found in Yoshioka et al. (2019), which we summarise here. The base model simulation was subject to a multi-year spin-up period. Parameter perturbations were then applied distinctly to individual ensemble members (which branch from the base model) and spun-up for a further month. We then ran each simulation for a further 12 months to produce the data used here. Horizontal winds and temperatures in the simulations are

nudged towards European Centre for Medium-Range Weather Forecasts (ECMWF) ERA-Interim reanalyses for 2008 between approximately 1.2 and 80 km using a 6-hour relaxation timescale. Nudging means that pairs of simulations have identical synoptic-scale features, which enables the effects of perturbations to aerosol and chemical processes to be quantified using single-year simulations, although the magnitude of forcing will vary with the chosen year (Fiedler et al., 2019; Yoshioka et al., 2019).

**2.2 Creation of perturbed parameter model variants**

Our method to determine observational constraint on the model parameters and radiative forcings involves producing a very large set of 'model variants', each with a different combination of parameter values, and then ruling out model variants for which a set of model outputs are judged to be implausible against measurements (see section 2.4). The model variants were generated using a perturbed parameter ensemble (PPE) of 235 model simulations of HadGEM3-UKCA (the 'AER PPE'

detailed in Yoshioka et al., 2019) that samples 26 sources of uncertainty in the aerosol model (Carslaw et al., 2017; Yoshioka et al., 2019) – see Table A1 in Appendix A.

A set of 235 simulations alone is much too small to allow statistical analysis of model performance across 26 dimensions of parameter space. We therefore built Gaussian Process emulators (surrogate models) using the PPE simulations as training data (Lee et al., 2011b), which define how the model outputs vary continuously over the 26-dimensional parameter space and enable

dense sampling over parameter uncertainty. Separate emulators were built describing the monthly mean value of each model output in each model grid cell. We then used Monte Carlo sampling from these emulators to produce output for a set of 1 million model variants (parameter input combinations). Uniform distributions were assumed for each parameter in this

sampling. The emulator is not a perfect representation of a model output, but its uncertainty can be estimated and accounted for in the model-measurement comparison. In the rest of this paper we refer to the emulator-derived values of model outputs at each sampled 26-d input combination as a 'model variant'.

The AER PPE samples only uncertainties in the aerosol component of the model and the radiative forcing does not account for atmospheric and cloud adjustments – i.e., it is a radiative forcing (RF) rather than an effective radiative forcing, which we analysed in previous papers (Johnson et al., 2018; Regayre et al., 2018). The prior (unconstrained) 95% credible interval of global mean aerosol RF is -2.23 ± 0.94 W m$^{-2}$. However, because of the way that multiple parameters compensate (Lee et al., 2016; Regayre et al., 2018), the forcing uncertainty in this PPE is similar to the AER-ATM PPE in which additional physical atmosphere model parameters were perturbed and cloud adjustments accounted for (Yoshioka et al., 2019). Because the AER PPE analysed here samples only aerosol uncertainties, we restrict the constraints to measurements of aerosol properties. In future work we will extend the analysis to radiation, precipitation and cloud measurements that are relevant to the wider range of parameters in the AER-ATM PPE.

The choice of the 26 perturbed parameters and their uncertainty ranges were defined using expert elicitation (Yoshioka et al., 2019). The parameters (Table A1; full descriptions given in Yoshioka et al., 2019) relate to natural and anthropogenic emission fluxes of aerosol precursor gases and primary particles, the properties of primary particles (size), aerosol processes, aerosol hygroscopicity, removal rates and cloud droplet formation (updraft speed). The list of parameters is not exhaustive, but one-at-a-time parameter perturbation tests were used to show that any other parameters have a smaller effect regionally and globally in our model than the set we chose. Finally, we note that the evaluated uncertainty in global annual mean RF in this study differs from that shown in Yoshioka et al. (2019) as we have used uniform parameter distributions when sampling over the parameter uncertainty space, while elicited parameter distributions were used in Yoshioka et al. (2019). Our choice to use uniform distributions here means that the constraint can be fully attributed to the model-measurement comparison.

## 2.3 Measurements

We use aerosol measurements from ground stations, ship campaigns and aircraft campaigns covering the following aerosol properties: aerosol optical depth (AOD), PM2.5 concentrations, sulphate mass concentrations, organic carbon mass concentrations, and number concentrations of particles larger than 3 nm dry diameter ($N_3$) and 50 nm dry diameter ($N_{50}$) – see Appendix B and Table S1 in the supplementary data file. All measurements used are from within the boundary layer, which we define to be at an atmospheric pressure greater than 800hPa. We do not attempt to constrain aerosol properties above the boundary layer.

The measurements were all made at specific locations and times (i.e. they are 'point measurements') in the period from October 1995 to December 2015, and we use measurements from all years within this period regardless of whether the year of the measurement matches the year of the PPE model simulations. (We take account of the inter-annual differences by incorporating





an error term in the constraint process, see Section 2.4). The measurements were aggregated to monthly mean values in grid cells of size 2.50° longitude by 3.75° latitude (4 model grid boxes of the N96 model grid). In cases where there is more than one measurement in a model grid cell, the observed values were averaged. This processing resulted in 9464 monthly-aggregated grid-box measurements (over 6 aerosol properties and 12 months). Figure 1 shows the global spatial coverage of the gridded measurements for each aerosol property, along with the monthly temporal coverage for each measurement, which is indicated by the colour-scale. Table 1 shows the breakdown of the number of grid-box measurements by variable and month.

The AOD data are level 2.0 (quality assured) monthly-mean data at 440 nm wavelength from the AERONET (Aerosol Robotic Network) network (Giles et al., 2019; Holben et al., 1998). Our dataset includes an average of 312 aggregated grid-box measurements for comparison in each month. Figure 1 shows that the measurements are well distributed across all continental regions except Antarctica. The coverage at high northern latitudes is relatively sparse, and there are only a small number of island measurement that are representative of marine aerosol environments. The temporal coverage is very good, with the majority of stations providing measurements in all months of the year.

The PM2.5 and sulphate concentration data come from multiple large networks. The sulphate concentration data are from the Interagency Monitoring of Protected Visual Environments (IMPROVE) network (USA), the European Monitoring and Evaluation Programme (EMEP) network and the Acid Deposition Monitoring Network in East Asia (EANET). For PM2.5, we use data from the IMPROVE network, the World Data Centre for Aerosols (WDCA) (European sites), the Asia-Pacific Aerosol Database (A-PAD) and the Canadian National Air Pollution Surveillance Program (NAPS). Other PM2.5 measurements are included from smaller networks and individual stations in Australia, South America, Taiwan and South Africa, as well as sulphate and PM2.5 data recorded at the Station for Observing Regional Processes of the Earth System (SORPES) in Nanjing, East China. The PM2.5 data (except for the SORPES site) were obtained, processed and gridded to the N96 model grid as described in Browse et al. (2019). Figure 1 shows that these PM2.5 and sulphate measurements are highly clustered over polluted land areas of the Northern hemisphere, mostly in North America and Europe with limited coverage elsewhere, especially in remote and marine areas. Nearly all stations in these data sets have full temporal coverage, leading to approx. 150 and 170 aggregated grid-box measurements for comparison in each month for PM2.5 and sulphate respectively (Table 1).

For $N_{50}$ particle concentrations and OC concentrations we have a mixture of measurements from a small number of land-based ground stations along with measurements taken over marine environments from ship and aircraft campaigns (see Table A2). The $N_{50}$ concentration data were mainly derived from size distribution measurements and gridded to the N96 model grid as described in Browse et al. (2019). The amount of campaign data, and hence global spatial coverage in the gridded data, is greater for $N_{50}$ than for OC (Figure 1), and the number of aggregated grid-box measurements is variable between months (Table 1). Due to the nature of field campaigns, the temporal coverage is much sparser for these variables, with each campaign only measuring for 1-3 months of the year, shown by the blue colours for the data of these variables in Figure 1.





The measurement data for $N_3$ particle concentration has the smallest number of grid-box measurements over the year and spatially is the sparsest data set included here. The data for this aerosol property comes from only 12/13 ground stations (ACTRIS; Asmi et al., 2013), which are mostly located in Europe, with one in the Arctic, one in Antarctica and one in Northern India. The $N_3$ concentrations at each site were derived directly by integrating size distribution measurements. This data was then averaged over multiple years for each month and location by the authors.

|       | AOD  | Sulphate | PM2.5 | OC  | $N_3$ | $N_{50}$ |
|-------|------|----------|-------|-----|-------|----------|
| **Jan**   | 294  | 149      | 168   | 6   | 13    | 77       |
| **Feb**   | 301  | 148      | 168   | 14  | 13    | 90       |
| **Mar**   | 309  | 151      | 170   | 82  | 13    | 148      |
| **Apr**   | 316  | 151      | 170   | 74  | 12    | 199      |
| **May**   | 322  | 149      | 167   | 23  | 12    | 64       |
| **Jun**   | 320  | 150      | 170   | 23  | 12    | 96       |
| **Jul**   | 323  | 148      | 172   | 23  | 13    | 115      |
| **Aug**   | 326  | 148      | 169   | 23  | 13    | 109      |
| **Sep**   | 321  | 147      | 166   | 22  | 13    | 133      |
| **Oct**   | 315  | 147      | 165   | 41  | 13    | 119      |
| **Nov**   | 309  | 146      | 168   | 37  | 13    | 155      |
| **Dec**   | 298  | 147      | 169   | 15  | 12    | 67       |
|       |      |          |       |     |       |          |
| **Total** | 3754 | 1781     | 2022  | 383 | 152   | 1372     |

**Table 1.** The number of monthly aggregated grid-box measurements for each variable in each month. The total number over all months and all variables is 9464.

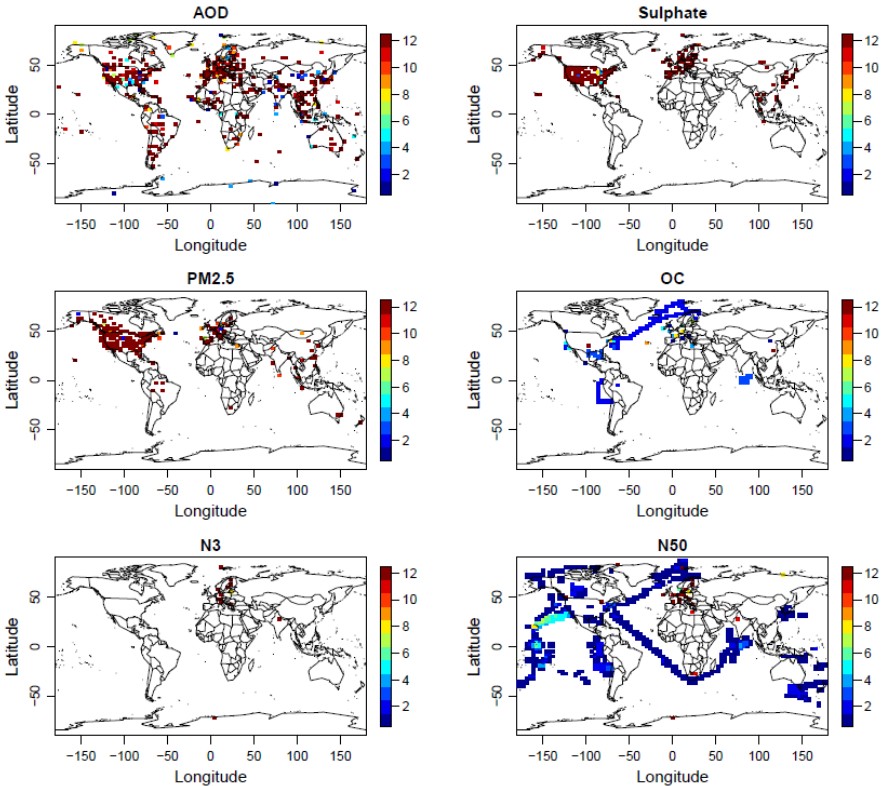

**Figure 1.** The distribution of measurements used in the constraint. The colours indicate the number of months covered by the measurements (although the data may not cover all days within a month).

### 2.4 Constraint methodology

We apply the statistical methodology of history matching, which has been applied to complex models in a range of fields, including epidemiological modelling of virus transmission (Andrianakis et al., 2017), risk assessment for oil field developments (Craig et al., 1997), modelling galaxy formation (Rodrigues et al., 2017) and climate modelling (Edwards et al., 2011; McNeall et al., 2016; Williamson et al., 2013). The methodology is described in detail in previous papers (Johnson et al., 2018; Regayre et al., 2018), which built upon our earlier study (Lee et al., 2016). We therefore describe the overall

methodology only briefly here, but present a full description of the new aspects related to using real measurements rather than 'synthetic' measurements (Johnson et al., 2018).

In the comparison of the model and measurements we account for emulator uncertainty, measurement uncertainty (instrument error), representativeness uncertainties (caused by spatial and temporal mismatches in resolution and sampling between model and measurements), and potential structural model uncertainty. The model-measurement difference together with these

15 measures of uncertainty is incorporated into an 'implausibility measure' and our model constraint procedure in order to identify implausible parts of parameter space (model variants).



### 2.4.1 Implausibility measure

The implausibility metric $I(x)$ is calculated for each of the 1 million model variants $x$, for each gridded measurement. $I(x)$ weights the difference between the model and measurements by the uncertainties associated with the comparison (Craig et al., 1996; Williamson et al., 2013):

$$I(x) = \frac{|M - O|}{\sqrt{[Var(M) + Var(O) + Var(R) + Var(S)]}}, \tag{1}$$

where $M$ is the estimate of model output calculated using the emulator and $O$ is the observed value (the measurement). In the denominator $Var(M)$ is the variance in the model estimate (associated with replacing the model with the emulator), $Var(O)$ is the variance in the measurement (i.e., instrument or retrieval uncertainty), $Var(R)$ is the variance associated with the comparison of the model with the measurements, called the representativeness error (Schutgens et al., 2017, 2016a, 2016b), and $Var(S)$ is a model structural error term.

A low value of the implausibility metric indicates either the model-measurement difference is small (i.e., the model is skilful) or that the uncertainty in the denominator is large (i.e., we cannot tell whether the model is skilful because the uncertainties are too large). Therefore, the implausibility metric allows model variants to be ruled out if the model-measurement difference is large and we can be confident that it is large.

The representativeness error $Var(R)$ has three components. $Var(R_{sp})$ ($sp$ = spatial) accounts for uncertainty associated with
spatial variability below the grid scale of the model, which means that a point measurement may not be representative of the grid-box mean (Schutgens et al., 2016b). $Var(R_{temp})$ ($temp$ = temporal) accounts for the temporal sampling of a measurement, which may not match the temporal sampling of the model (e.g. a ship track through the grid-box over a short time period which is compared with a monthly-mean model value (Schutgens et al., 2016a). $Var(R_{iav})$ ($iav$ = inter-annual variability) accounts for the fact that we sometimes match measurements and the model for the correct calendar month but not for the correct year.
This is necessary in cases where we use measurements from years for which we have not run the model. We assume that

$$Var(R) = Var(R_{sp}) + Var(R_{temp}) + Var(R_{iav}) \tag{2}$$

The magnitude of these errors is discussed in section 2.4.2.

The structural error term $Var(S)$ has been included in previous studies using the implausibility metric. It is intended to represent an estimate of the potential structural error in the model. Practically, however, we have no way to estimate this term for all variables at all times and geographical locations. We therefore set it to zero, and instead use very large values of
implausibility to point us towards potential structural errors in the model-observation comparison and constraint procedure, as described in section 2.4.3.



### 2.4.2    Estimation of uncertainty terms

Our estimates of the uncertainty terms in Equation 1 are preliminary and are designed to test our approach. We discuss in the conclusions the need to refine our understanding of these uncertainty terms.

For all aerosol properties we assume an instrument uncertainty of 10%, a spatial co-location uncertainty of 20%, and a temporal sampling uncertainty of 10% on the measured value. The spatial sampling uncertainty for monthly mean aerosol properties is estimated based on Schutgens et al. (2017, 2016b). These studies examined a typical spatially heterogeneous continental environment where the sampling error is dominated mainly by local aerosol sources that are not resolved by the global model. The magnitude of uncertainty is likely to vary globally (especially between land and ocean), with surface measurements

typically having larger errors than column measurements and the magnitude of error also depending on the location of a ground site with respect to the grid-box centre, but we do not account for these variations. We base our estimate of the temporal sampling uncertainty on Schutgens et al. (2016a) who quantified the error associated with the different temporal sampling of models and measurements (e.g., daily measurements or temporally sporadic measurements versus monthly mean model, etc.). The emulator uncertainty is taken from the Gaussian error on the emulator mean prediction, which is known for every

parameter combination (i.e., each of the 1 million model variants).

The inter-annual uncertainty was defined to be the standard deviation of monthly mean aerosol properties in each grid cell over a 30-year period. We take information from an analysis of the trend and variation of gridded aerosol properties in a HadGEM3-UKCA hindcast simulation over the period of 1980-2009 (Turnock et al., 2015). For each month and grid-box, the

monthly mean output of the aerosol variable of interest for each year of the simulation was obtained. These values were de-trended using linear regression and the resulting residuals were then analysed. We use a relative measure of monthly mean uncertainty defined by the standard deviation of these residuals divided by the de-trended mean. As an example, Figure 2 shows the relative standard deviation for the surface-level $N_{50}$ concentration in July.





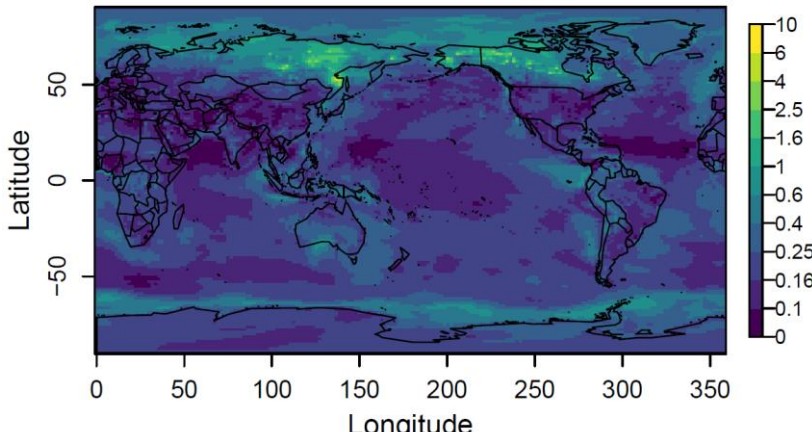

**Figure 2.** The relative standard deviation for surface-level $N_{50}$ conc. in July, used in the estimation of the inter-annual variability component of representativeness error $R_{iav}$.

### 2.4.3 Methodology for ruling out observationally implausible parts of parameter space

There is an element of subjectivity involved in comparing a model with point measurements and reaching a conclusion about the fidelity of the model. The comparison may indicate either: a) the model seems to be structurally adequate, but the parameters need to be adjusted to optimise agreement, or b) the model is structurally deficient (i.e., there are missing or incorrect process representations in the model). Structural deficiencies may be apparent, for example, because the model skill is particularly poor in one region or at one time of year, or it is not possible to obtain good skill across multiple variables simultaneously.

Our use of 1 million model variants and more than 9000 monthly aggregated grid-box measurements means that we need to automate the model-measurement comparison processes and detection of potential structural errors while also using the measurements to rule out implausible parts of parameter space. The difficulties for us in detecting structural errors are: a) we cannot inspect each of the 1 million model variants individually, so we need to rely on summary statistics; b) many of the aerosol point measurements are spatially and temporally sparse, so we cannot easily detect spatial and temporal changes in model skill that might indicate structural error; c) the measurements do not have the same spatial distribution in all months (because of brief, localised field campaigns) so spatial-temporal biases are hard to detect; d) the uncertainty in each measurement (particularly the representativeness error, section 2.4.2) is spatially and temporally heterogeneous and often very poorly defined.

Our approach is summarised in Figure 3. It is designed to rule out implausible parts of parameter space while avoiding doing so in cases where the biases shared by many model variants could be caused by structural errors in the model.



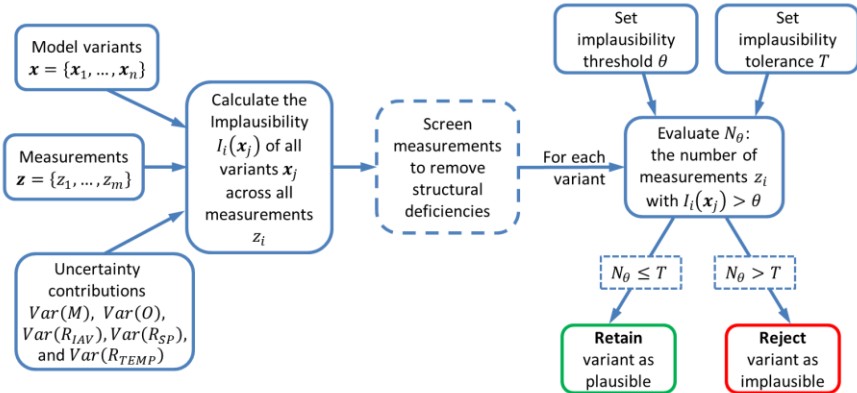

**Figure 3.** Flow chart detailing the process followed for each model variant $x_j$, in using the calculated implausibility over a set of m measurements $z = \{z_1, z_2, \cdots, z_m\}$ (for a single output variable $y$) simultaneously to constrain the model uncertainty.

The steps are:

1) The implausibility is quantified for each of the 1 million variants across all measurements of a single type in a particular month. Figure 4 shows an example for the measurements of $N_{50}$ in July. For each measurement (numbered on the horizontal axis, left plot), the distribution of the implausibility over the variants is shown by the bar representing the 95% credible interval.

2) Measurements are identified for which 97.5% of the model variants have an implausibility $I > 1$. These measurements are excluded from the constraint procedure (shown in red in Figure 4). We assume that this large implausibility for the significant

majority of variants indicates either there is a structural error in the model or that the model is unable to represent these point measurements because of its low spatial and temporal resolution (section 2.4.1). We flag these measurements for further investigation of potential structural errors or underestimated error terms (these are not examined further in this study).

3) Using all other measurements, (where more model variants have lower implausibility, shown in blue in Figure 4), we use the implausibility metric values to decide whether to rule each variant out as implausible, or retain it as plausible. If we ruled

out all model variants with high implausibility for each measurement in turn (treating the measurements independently, as in many emergent constraint studies), we could end up ruling out all parts of parameter space. Our criterion is therefore to rule out a model variant if more than a defined fraction (or number) of the measurements (tolerance $T$) exceeds a defined implausibility threshold ($\theta$). For example, we might rule out a model variant if more than 20% of measurements exceed an implausibility of 3.5 (i.e., bias is 3.5 times the expected error).

We apply this approach to the set of measurements for each variable (measurement type) in each month and then combine the constraints to a joint constraint over months and/or over variables such that if a variant is ruled out for any single month/variable combination, then it is also ruled out in the joint constraint. This method allows us to identify the set of model variants that capably represent measurements of a range of variables and across multiple locations and seasons. We extensively explored

various choices of the tolerance and threshold values in each variable/month case and found that the final constrained parameter ranges were reasonably robust, except when the number of measurements was small.

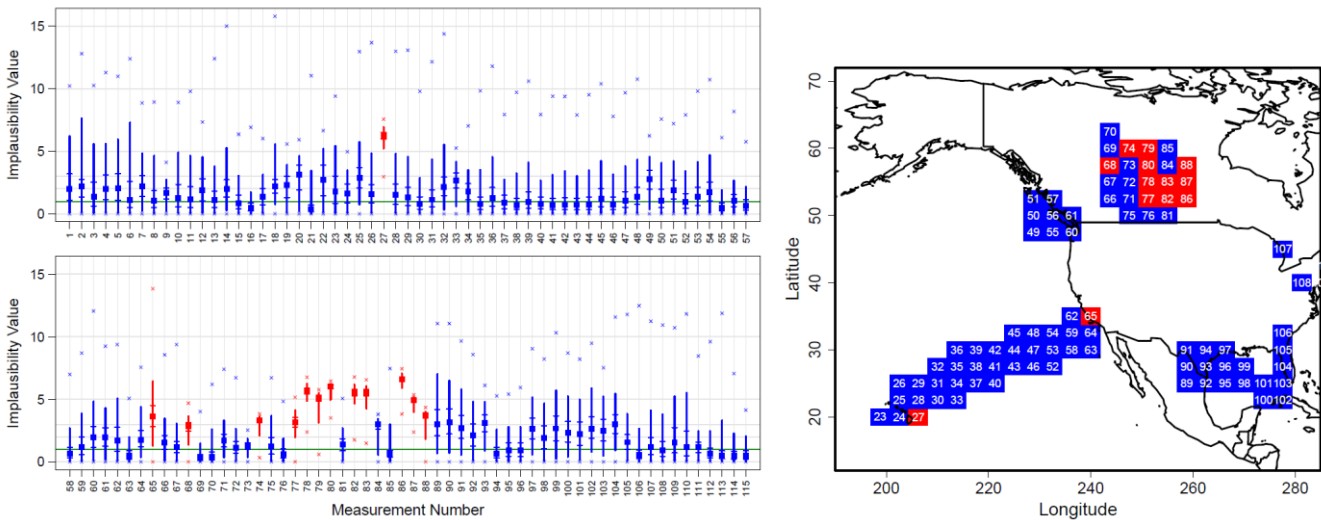

**Figure 4.** Left: The distribution of implausibility calculated over the 1 million model variants for each measurement in the July $N_{50}$
concentration set, shown vertically. For each measurement, numbered along the x-axis, the range of the implausibility distribution is shown by the outer crosses, the bar corresponds to the 95% credible interval (2.5% to 97.5% empirical quantiles), the horizontal markers through the bar show the inter-quartile range, and the square point is the median implausibility. Here we assume no structural error term in the implausibility calculations and use the implausibility distribution to identify potential structural errors. Measurements coloured red are ruled out as potential structural error cases (as the lower 95% credible interval bound is >1), and those coloured blue are retained and used in our constraint procedure. Right: Corresponding map to show the locations of the rejected July $N_{50}$ measurements (red) and those retained for constraint (blue), over the North Pacific and North America region (outside this region, all measurements were retained). We hypothesise that the red points over the Pacific correspond to ports with localised pollution sources while the red points over Canada correspond to localised fire emissions that are not represented at the resolution of the model.

Our choices of the threshold and tolerance for each measurement type are given in Table A2 in Appendix A. A wide range of values were tested in each case, starting with a set threshold of $\theta = 3.5$ and iterating through increasing tolerances $T$ up to a maximum of $T = 33\%$ (1/3 of the measurements), before further increasing $\theta$ by 0.5 (to a maximum of $\theta = 4.5$) and re-iterating over $T$ in order to retain (approximately) a chosen percentage of model variants. Approximately the same percentage of variants was attained for all months of a variable type and combined for an 'all months' constraint. Our final choices for each variable type on its own (left column in Table A2) were relaxed for the joint all-variables-months constraint (retaining a larger percentage of variants in each month for each variable, so a weaker constraint; right column in Table A2), in order to retain a reasonable number of model variants and avoid over-constraining on any one observational type.

Our assumption of zero structural error ($Var(S) = 0$) in the implausibility calculations means that structural errors in the model can easily come to light in our constraint process. This occurs either in the calculated implausibility values for a measurement (where large values are consistently produced over the 1 million variants covering the model uncertainty,





indicating a large model-measurement discrepancy, e.g. Figure 4), or when bringing together the constraint effects of different sets / types of measurements (where very few, if any, model variants that lead to plausible model output in all cases/measurement types simultaneously can be identified and retained). Even though we do not directly account for structural errors in the implausibility measure itself, our constraint approach offsets the effects of such errors on the achieved constraint

as much as possible. This is accomplished by screening out observations with large model-measurement discrepancies from the constraint process (step 2; Figure 4) and by relaxing the constraint criteria for the joint all-variables-months constraint. Through this approach we are able to produce as robust a constraint as possible, given the limitations we have in specifying structural and representational errors.

## 2.5  Interpretation of constrained parameter probability distributions

Observationally plausible parts of parameter space exist in 26 dimensions. We show the results as *1-dimensional marginal probability distributions*, which are 1-dimensional projections of the 26-dimensional parameter probability distribution. Figure 5 shows an idealised representation for a 2-dimensional parameter constraint. The white parts of the joint distribution are ruled out, leaving the shaded region of joint parameter space as observationally plausible. The effect on the marginal probability distribution of parameter 1 is to entirely rule out the lowest and highest values (i.e., there is no combination of these values of

parameter 1 with parameter 2 that produces an observationally plausible model). Where some values of parameter 1 are ruled out over the range of parameter 2, the likelihood of parameter 1 having those values is reduced.

In the results below, the parameter probability distributions therefore reflect the relative likelihood of the parameter having particular values, with lower probabilities indicating that there are fewer ways in which the parameter can be combined with the other 25 parameters to produce a plausible model. For conciseness in the results section we say, for example, that "a

measurement constrains the parameter to low values", which means that we retain a larger proportion of model variants with low values.

Figure 5 also shows the separate and joint effects of two observational constraints. We show this conceptually because it arises in the results. Measurement 1 rules out the lowest values of parameter 1 and suggests that parameter 1 is likely to be at the high end of the sampled range. Conversely, measurement 2 suggests that parameter 1 is likely to be at the low end of the range.

However, the correct interpretation of this situation is that intermediate values of the parameter are consistent with *both* measurements (measurement 1 is consistent with the model for all but the lowest values of the parameter and measurement 2 is consistent for all but the highest values). Only in cases where the two separate constrained parameter pdfs do not overlap can we conclude with certainty that there is likely to be a structural deficiency in the model. However, to obtain multivariate constraint we prevent this happening by screening out measurements with large model-measurement discrepancies and

relaxing the constraint criteria with each measurement type.



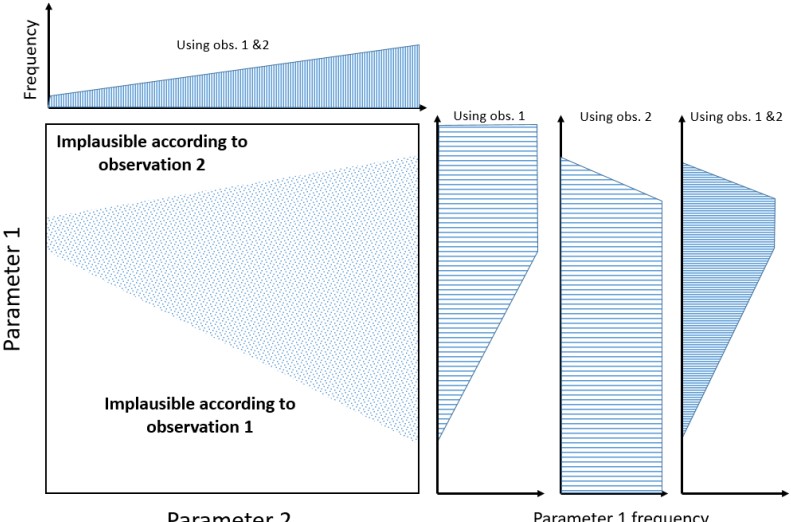

**Figure 5.** Schematic of parameter constraint in two dimensions using two measurements.

## 3 Results

### 3.1 Constraint using individual measurement types

Figure 6 shows the constrained marginal parameter distributions for all parameters based on using individual measurement types (each column on left) and all measurement types together (right column).

**AOD measurements** constrain aerosol and precursor emissions to low values and removal rates to high values. These constraints imply that the PPE produces generally too-high AODs across the sampled parameter space, which is the case (section 3.4). In particular, sea spray emissions higher than about 3.6 times the baseline emissions are ruled out, but emissions

down to as low as 0.125 times the baseline emissions are plausible. For anthropogenic $SO_2$ emissions, the likelihood of the emissions scale factor being below 1 (corresponding to the default value from the inventory) increases from 55% to 70% on constraint. For BVOC emissions, the likelihood of the emissions (or effectively the production of SOA) being more than 3 times the default emission of 46 Tg/y (=138 Tg/y) is reduced from 31% to 13%, but all lower values from the default emission down to our lower bound of 37 Tg/y are equally plausible.

The AOD measurements also constrain the model to low values of other parameters: more variants with higher cloud droplet pH values are ruled out (judged implausible) and as a result cloud droplet pH is nearly 3 times as likely to be below the central value of its range of 5.8 as above it, which is consistent with a higher likelihood of slower production of sulphate aerosol from in-cloud $SO_2$ oxidation. The hygroscopicity of OC in the particles ($\kappa_{OC}$) is also weakly constrained to low values, which reduces the water content of aerosol and reduces AOD. The rate of aerosol scavenging by precipitating raindrops (the Rain_Frac

parameter) is weakly constrained to high values.




**Sulphate measurements** strongly constrain $SO_2$ emissions to low values, which is consistent with the AOD constraint. Given this constraint, the $SO_2$ emissions have a 78% likelihood of being below the default value from the inventory and the median emission is reduced to 0.78 times the default. Also consistent with the AOD constraint, the deposition rate of accumulation mode particles is constrained strongly to high values, with an 87% likelihood of the rate being above the default value.

Likewise, the $SO_2$ dry deposition rate is constrained fairly weakly to higher values, with a 60% likelihood of the scaled value being above the default value. Each of these constraints is consistent with too-high sulphate concentrations in many of the sampled model variants across the parameter uncertainty space (section 3.4).

**PM2.5 measurements** have a similar effect to AOD on some parameters, but for others there are differences. Emissions of sea spray and BVOC emissions are constrained similarly (to low values). However, $SO_2$ emissions and cloud droplet pH are

weakly constrained to higher values and the dry deposition rate of accumulation mode particles is weakly constrained to low values, opposite to the AOD and sulphate constraints for these parameters. The PM2.5 measurements also weakly constrain the residential combustion emissions to high values. PM2.5 and AOD are strongly correlated in the PPE (Johnson et al., 2018), so differences in the constrained parameters most likely reflect differences in the spatial distribution of the measurements (Figure 1) and how that maps on to the spatial variations in sensitive parameters. As described in section 2.5, these apparently

opposing constraints are not necessarily inconsistent: for AOD and PM2.5 there may be other parameter settings that can be combined with low $SO_2$ emissions to achieve agreement with the measurements (so the space is not ruled out).

**Organic carbon (OC) measurements** strongly constrain the scaled magnitude of residential carbonaceous emissions to a narrower credible interval of about 0.3-1.8 centred near the default value specified in the emissions. Emissions above 2.0 times the default value are effectively ruled out and there is only a 13% likelihood of the emissions being below half the default

value. Fossil fuel emissions have a 70% likelihood of being above the default emission value. The OC measurements also constrain the scaled BVOC emissions in a similar way to PM2.5 and AOD, with scaled emissions above about 2.1 times (97 Tg/y) having only a 31% likelihood (compared to 50% prior to constraint). OC measurements also constrain the lowest values of BVOC emissions, which was not achieved with PM2.5 and AOD. The likelihood of the scaled emissions being below 1 (46 Tg/y) is 6% (compared with 11%). The dry deposition rate of Aitken mode particles is constrained to the low part of the range,

which will tend to increase OC concentrations in the atmosphere consistent with the constraint of fossil fuel emissions to high values. There is also a weak constraint of the ageing rate towards higher values, which has a 55% likelihood of being in the upper half of the range. The rate of aerosol scavenging by precipitating raindrops (Rain_Frac parameter) is constrained similarly, but to lower values. Again, although weak, these two constraints imply slower ageing, slower removal rates, longer OC lifetime and higher atmospheric concentrations. Biomass burning emissions are only very weakly constrained towards

lower emissions. The lack of constraint on the biomass burning emissions from OC measurements here is likely a result of the limited coverage, if any, of the OC measurements in regions important for biomass burning such as Africa and South East Asia (Figure 1).

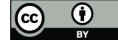



**Figure 6.** Marginal parameter distributions after constraint using individual measurement types over all months (6 columns on the left) and after using all measurement types over all months together (right column). The 25th, 50th and 75th percentiles of each constrained distribution are shown in the central boxes, and the parameter values on the x-axes correspond to values as they are used in the model [parameters that are multiplicative scaling factors are shown on the $\log_{10}$ scale], covering the full parameter ranges (Yoshioka et al., 2019). The corresponding choices of threshold $\theta$ and tolerance $T$ that were applied in the constraint process to generate these results are given in Table A2 (left column for each individual measurement type; right column for the joint measurement-types constraint), along with the percentage of model variants that is retained in the constrained sample in each case. See section 2.5 for a definition of marginal parameter distributions.

**Particle concentration ($N_3$ and $N_{50}$) measurements** constrain a wider range of parameters than the measurements of mass-related properties. The rate of boundary layer nucleation is strongly constrained to the low part of the sampled range by the $N_3$ measurements (a 77% likelihood of being below the default rate), suggesting $N_3$ concentrations are generally too high across the PPE. $N_3$ also weakly constrains the dry deposition of Aitken and accumulation mode particles to low values. Low deposition rates of accumulation mode particles (hence higher atmospheric concentrations) will result in a higher condensation sink and





more removal of sulphuric acid that participates in particle nucleation, so this is consistent with the constraint of nucleation rates to low values. The constraint of Aitken mode deposition to low values is less obvious. Aitken mode particles can contribute substantially to $N_3$, so low deposition rates would enhance $N_3$ (opposite to the constraint on nucleation rates). However, nucleation rates are constrained to very low values, so in such a situation Aitken particles can begin to act as a sink

term for nucleation by affecting the condensation sink and by growing into accumulation mode particles. BVOC emissions are not constrained by $N_3$ measurements, even though SOA enters into the nucleation rate expression. This is most likely because high BVOC emissions also enhance total SOA, which acts as a condensation sink for nucleation, so the two effects cancel (Carslaw et al., 2013b).

For $N_{50}$, the constraints are consistent with shifting the $N_{50}$ concentrations in the ensemble towards lower values (section 3.4).

$N_{50}$ has very little effect on the range of boundary layer nucleation rate. In contrast, a previous study found that boundary layer nucleation made a statistically significant difference to model skill at about half of the ground sites they analysed (Reddington et al., 2011) – although that study tested the effect of including or not including boundary layer nucleation rather than perturbing the rate as we do here. Without boundary layer particle formation the model was structurally deficient and had poor skill at around half the sites analysed. However, our results show that uncertainty in the parameter value itself is unimportant

when other parameter uncertainties are considered. This parameter is unconstrained by $N_{50}$ measurements because there are many alternative ways of achieving model-measurement agreement.

$N_{50}$ measurements also tend to constrain primary particle emissions to the lower end of the range (fossil fuel and primary sulphate emissions), albeit weakly. Residential particle emissions are not constrained, but the measurements we used are not well located to achieve this. It also constrains the emitted particle diameters to the high end of their ranges (fossil fuel, primary

sulphate), which is again consistent with low number concentrations (since we perturb emission diameter independently of the mass, so number concentration is affected). The constraint of particle emission sizes is consistent with a previous study that showed CCN concentrations are sensitive to the assumed size (Reddington et al., 2011). Our results show that $N_{50}$ measurements allow the emission size to be constrained, even though there are many other compensating factors that can affect CCN concentrations. $N_{50}$ weakly constrains cloud pH to higher values, consistent with greater production of sulphate aerosol

and a higher sink for nucleation. BVOC emissions are constrained to the low end, which is consistent with reduced growth of nucleation mode particles into the Aitken and accumulation modes. $N_{50}$ also constrains depositions rates: accumulation mode deposition is constrained to low values and Aitken mode deposition to high values, suggesting a shift in the aerosol size distribution towards larger aerosols is consistent with $N_{50}$ measurements.

## 3.2 Seasonal variations in constraint

Many of the parameter constraints vary seasonally, which can be linked to seasonal variations in emissions and parameter sensitivity. Some examples are shown in Figure 7. Cloud pH is constrained more by AOD in winter (Figure 7a) when in-cloud oxidation of $SO_2$ by ozone dominates sulphate production. BVOCs are constrained by AOD only in northern-hemisphere





summer when the emissions are strong (Figure 7b). There are several other seasonal variations in the constraint effect from AOD measurements that we do not show. For example, anthropogenic $SO_2$ emissions are constrained by AOD more in winter because the AOD uncertainty in summer is dominated by the uncertainty in SOA. The hygroscopicity of OC is also constrained more in summer when OC is a larger component of the aerosol. Biomass burning emissions are constrained in NH summer as

expected from wildfire emission seasonality and the northern-hemisphere bias of our measurements dataset. Residential emissions are only constrained in winter when emissions are high. Microphysical process rates (dry deposition of accumulation mode and wet scavenging rates) are consistently constrained throughout the year.

For PM2.5, the seasonality of constraint is very similar to AOD with one notable exception. The dry deposition rate of accumulation mode particles is constrained to high values in summer (consistent with AOD and sulphate), but to low values

in the winter (Figure 7c). This may occur just because of the way in which the combinations of parameters control PM2.5 – for example BVOCs can account for PM2.5 in summer so high dry deposition rates cannot be ruled out. However, it may also indicate a structural deficiency, with the low deposition rates in winter implying that PM2.5 has missing sources in winter but not in the summer, such as nitrate.

For $N_3$, we find that the boundary layer nucleation rate is constrained only in summer when photochemical production of the

nucleating vapours is fast (Figure 7d). This is consistent with previous studies that have examined the seasonal cycle of organic-mediated nucleation (Riccobono et al., 2014). Similarly, $N_3$ measurements constrain $SO_2$ emissions and cloud droplet pH in summer when nucleation is most active. This is in contrast to the AOD and sulphate measurements, which constrained these two parameters in winter when their relative contribution to aerosol mass is greater.

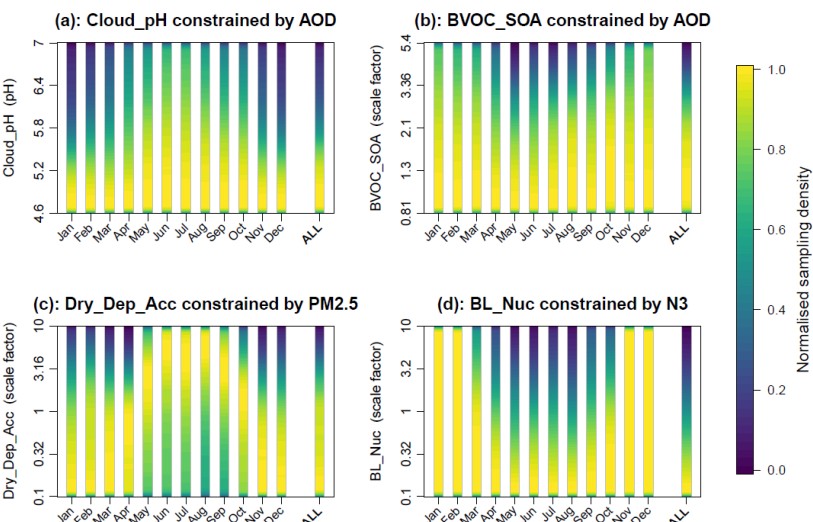

**Figure 7.** Seasonal variation in the constraint of parameter marginal probability distributions. The examples are a) constraint of the pH of cloud droplets (Cloud_pH) parameter using global AOD measurements, b) constraint of SOA production from BVOCs (BVOC_SOA) using AOD measurements, c) constraint of the dry deposition rate of accumulation mode particles (Dry_Dep_Acc) using global PM2.5 measurements, d) constraint of boundary layer nucleation rates (BL_Nuc) using $N_3$ measurements mainly over Europe



For $N_{50}$ we find that parameter constraints do not vary smoothly through the year (not shown). This is because the $N_{50}$ measurements we have used are primarily from campaigns, which move around the globe, resulting in constraint of regionally important parameters. This is one indication that we need to add long-term network measurements of $N_{50}$ to the dataset.

### 3.3 Constraint using all measurement types

The multi-variate constraint is shown as the right-hand column of pdfs in Figure 6 and Table 2 shows corresponding parameter distribution statistics from this constraint. For each individual variable / month constraint that feeds into this multi-variate constraint, the implausibility threshold and tolerance criteria ($\theta$ and $T$), were relaxed from the individual measurement constraints to retain approximately 75% of the 1 million model variants (Table A2). This relaxed criteria leads to measurements that provide stronger constraint being down-weighted and individual parameter constraints becoming weaker, but it means that we are able to avoid over-constraining on any one measurement type. Using all measurement types together leads to retention of only 2.1% of the original 1 million model variants as plausible models (nearly 98% rejected; Table A2). In most cases the marginal parameter distributions from this constraint can be understood in terms of the combination of individual constraints described above.

**Boundary layer nucleation rates** are constrained to the low end of the range, which can be attributed almost entirely to the $N_3$ measurements. However, the constraint is slightly weaker than when just $N_3$ measurements are used because of the need to relax the tolerances and thresholds applied when ruling out model variants using multiple measurement types (section 2.4.3). The nucleation rate is constrained such that the likelihood of it being in the lower half of the range (0.1-1 times the default value) is 70% - more than twice the likelihood of it being in the upper half of the range (1-10 times the default value).

**The pH of cloud droplets**, which controls aqueous-phase oxidation of $SO_2$ to form sulphate aerosol, is constrained to be more likely in the middle of our elicited range. This results from a combination of AOD and sulphate measurements constraining it to the lower end of the range and PM2.5 measurements constraining it to the higher end. Observational constraint is unable to rule out any of the pH values between 4.6 and 7.0, although there is reduction of 0.13 in the 95% credible interval to 4.69-6.84 (from 4.66-6.94 before constraint) and a larger reduction of 0.32 in the interquartile range to 5.24-6.12 (from 5.2-6.4 before constraint).

**Biomass burning emissions** are weakly constrained. The likelihood of emissions being more than a factor 2 above the default value is reduced to 14% (from 25%), but all values below this down to 0.25 times the default value are still equally likely, as they were before constraint.

**Residential carbonaceous emissions** are constrained primarily through a combination of PM2.5 and OC measurements. This emissions scaling parameter is constrained to be most likely near the middle of its range around the default setting, with emissions higher than about 2.7 times the default emission rate ruled out completely and also some weaker constraint at the lower end of the range. The 95% credible interval has significantly shifted towards lower values, from 0.27-3.73 times the



default value before constraint to 0.27-1.85 times the default value after constraint, with the constrained interquartile range being 0.46-1.06 (Table 2).

**The diameter of fossil fuel particles** is constrained mainly through the $N_{50}$ measurements towards larger diameters, with a likelihood of being in the upper half of our elicited range (60-90 nm diameter) of 61% and the median of this parameter distribution shifting to a larger diameter on constraint, increasing from 60nm to 65.63nm

**Sea spray emissions** are constrained through a combination of AOD and PM2.5 measurements, and to a lesser extent by $N_{50}$. The multivariate constraint is slightly weaker than was achieved by AOD and PM2.5 individually, although we are still able to rule out emissions in the range 4.7-8 times the default value. Emissions in the range 0.125-2.8 times the default value are not strongly constrained by any of the measurements.

**Anthropogenic $SO_2$ emissions** are strongly constrained to the lower part of the elicited range by a combination of AOD and sulphate measurements. The emissions are most likely to be at the lower end of our elicited range (0.6 times the default value) and the likelihood of the emissions being in the range 0.6-1 times the default value is 82%. Our interquartile range after constraint is 0.67-0.93 times the baseline emission value of 98 Tg/y from the MACCity inventory, so 65-91 Tg/y. Our constrained range therefore lies largely below the baseline value, with only an 18% probability of it being above the baseline value. Liu et al. (2018) have developed a new $SO_2$ emission inventory based on OMI measurements. They did not provide a global estimate of $SO_2$ emissions, but over the US and Europe, where most of our sulphate measurements are located, their revised emissions are 40% lower than in HTAP, which is in the same direction as our constraint. In their inverse model study, Lee et al. (2011a) estimated global land $SO_2$ emissions of 100-105 Tg/y (with an estimated uncertainty of 20%), in agreement with MACCity emissions, but their central value is around our 85[th] percentile.

**BVOC emissions** are constrained to a central value that corresponds to a global annual SOA production of about 86.5 Tg/y. No values in the parameter range (corresponding to an emissions range of 37-250 Tg/y) are ruled out, although the likelihood of SOA production being in either the upper (above 150Tg/y) or lower (below 60Tg/y) quadrants of the scaling range is significantly reduced and the interquartile range of the parameter distribution has reduced from 60-155 Tg/y to 62-111 Tg/y. BVOCs were constrained in Spracklen et al. (2011) using global Aerosol Mass Spectrometer measurements (which we also used) and a set of model runs that perturbed combinations of biogenic monoterpene and isoprene emissions as well as an anthropogenic VOC. Here we have used a combination of AOD, PM2.5, OC, $N_{50}$ and $N_3$ measurements, all of which are influenced by SOA. Their best estimate of the global SOA source was 140 Tg/y with an uncertainty range of 50–380 Tg/y. This included 100 Tg/y from anthropogenic sources (which they called anthropogenically controlled SOA), which we do not include in our set of perturbed parameters. When we use just global OC from AMS measurements we find a 95% range on BVOC SOA of 42-195 Tg/y. Measurements of PM2.5, AOD and, to a lesser extent $N_{50}$, provide additional constraint, resulting in a 95% interval of 40-172 Tg/y and a median of 86.5 Tg/y. This range accounts for potential compensating effects of uncertainty in deposition rates and other parameters that were not considered in Spracklen et al. (2011).



**The dry deposition rate of Aitken mode particles** is weakly constrained to low values, which comes mainly from the OC and $N_3$ observational constraint. The likelihood of the deposition rate being in the range 0.5-1.0 times the default value (1.0) is increased from 50% to 60% on constraint.

**The dry deposition rate of accumulation mode particles** is constrained to the middle of the range. This is likely because sulphate measurements constrain the deposition rate to be towards the high end while the other measurements constrain it towards the low end. The multivariate constraint is weaker than when individual measurement types are used (AOD, sulphate, PM2.5, $N_{50}$, $N_3$), which results from relaxing the individual constraints in order to retain a reasonable number of model variants when multiple variables do not agree on the best value of the deposition rate.

**The dry deposition rate of SO₂** is constrained to the upper part of the elicited range, with the likelihood of it being in the range 1-5 times the default value (i.e., an increase in SO2 emissions) now 62% after constraint.

| Parameter | Median | | 95% Credible Interval | | 95% CI Range Ratio (Constrained/Unconstrained) |
|---|---|---|---|---|---|
| BL_Nuc* | 1.00 | 0.47 | (0.11 , 8.91) | (0.11 , 6.79) | 0.94 |
| Ageing | 5.15 | 5.51 | (0.54 , 9.76) | (0.55 , 9.81) | 1.00 |
| Acc_Width | 1.50 | 1.50 | (1.21 , 1.79) | (1.21 , 1.79) | 1.00 |
| Ait_Width | 1.50 | 1.55 | (1.21 , 1.79) | (1.23 , 1.79) | 0.97 |
| Cloud_pH | 5.80 | 5.67 | (4.66 , 6.94) | (4.69 , 6.84) | 0.94 |
| Carb_FF_Ems* | 1.00 | 1.01 | (0.52 , 1.93) | (0.52 , 1.93) | 1.00 |
| Carb_BB_Ems* | 1.00 | 0.83 | (0.27 , 3.73) | (0.26 , 3.28) | 0.97 |
| Carb_Res_Ems* | 1.00 | 0.73 | (0.27 , 3.73) | (0.27 , 1.85) | 0.73 |
| Carb_FF_Diam | 60.00 | 65.63 | (31.50 , 88.50) | (35.16 , 88.80) | 0.94 |
| Carb_BB_Diam | 195.00 | 194.97 | (95.25 , 294.75) | (94.89 , 295.27) | 1.00 |
| Carb_Res_Diam | 295.00 | 299.73 | (100.25 , 489.75) | (99.26 , 492.03) | 1.01 |
| Prim_SO4_Frac* | $3.16 \times 10^{-4}$ | $2.41 \times 10^{-4}$ | $(1.33 \times 10^{-6} , 7.50 \times 10^{-2})$ | $(1.26 \times 10^{-6} , 7.46 \times 10^{-2})$ | 1.00 |
| Prim_SO4_Diam | 51.50 | 56.43 | (5.43 , 97.58) | (7.06 , 98.04) | 0.99 |
| Sea_Spray* | 1.00 | 0.82 | (0.14 , 7.21) | (0.14 , 3.69) | 0.83 |
| Anth_SO2* | 0.95 | 0.77 | (0.61 , 1.47) | (0.61 , 1.35) | 0.90 |
| Volc_SO2* | 1.30 | 1.25 | (0.73 , 2.31) | (0.73 , 2.30) | 1.00 |
| BVOC_SOA* | 2.09 | 1.88 | (0.85 , 5.15) | (0.86 , 3.74) | 0.82 |
| DMS* | 1.00 | 0.97 | (0.52 , 1.93) | (0.52 , 1.92) | 1.00 |
| Dry_Dep_Ait* | 1.00 | 0.88 | (0.52 , 1.93) | (0.51 , 1.90) | 1.00 |
| Dry_Dep_Acc* | 1.00 | 0.76 | (0.11 , 8.91) | (0.11 , 5.73) | 0.90 |





| | | | | | |
|---|---|---|---|---|---|
| Dry_Dep_SO2* | 1.00 | 1.45 | (0.22 , 4.61) | (0.23 , 4.76) | 1.00 |
| Kappa_OC | 0.35 | 0.36 | (0.11 , 0.59) | (0.11 , 0.59) | 1.00 |
| Sig_W | 0.40 | 0.40 | (0.12 , 0.68) | (0.11 , 0.69) | 1.04 |
| Dust* | 1.00 | 1.03 | (0.52 , 1.93) | (0.52 , 1.94) | 1.00 |
| Rain_Frac | 0.50 | 0.50 | (0.31 , 0.69) | (0.31 , 0.69) | 1.00 |
| Cloud_Ice_Thresh | 0.30 | 0.29 | (0.11 , 0.49) | (0.11 , 0.49) | 1.00 |

**Table 2.** Marginal parameter distribution statistics (median and 95% credible interval) for the unconstrained sample of 1 million model variants in black and the constrained sample of model variants from the constraint using all measurement types simultaneously in red. (*Parameter values given as a multiplicative scaling). The final column shows the ratio of the constrained to the unconstrained 95% credible interval range, accounting for the nature of the parameter (absolute or multiplicative) by using the $\log_{10}$ scale for the calculation when the parameter is a multiplicative scaling.

## 3.4 Model-measurement comparison

Figure 8 compares the unconstrained (black) and constrained distributions of model outputs with the measurements (green). We show the results when single measurement types are used for constraint (blue) and when all measurement types are used together (red). The constraint procedure clearly rules out wide ranges of model outputs that are inconsistent with the measured values, shown by the vertical green lines. For example, the unconstrained distribution of mean global sulphate concentration (at the measurement sites) extends up to about 6 µg m$^{-3}$ in January, but the tail of the distribution is limited to 3 µg m$^{-3}$ after constraint.

The constrained model distribution sometimes agrees much better with the measurements when only a single measurement type is used compared to when all measurements are used. The weaker multi-variate constraint is because we relax the constraint on individual variables so as not to rule out all model variants. This effect is most apparent for sulphate and PM2.5. The mean of the constrained PM2.5 distribution using all measurements is about 40% lower than the mean of the measurements in January but the mean of the sulphate distribution is about 50% higher than the mean of the measurements. This is likely to indicate a structural error in the model that prevents good model-measurement agreement with both quantities in the same parts of model parameter space. One explanation could be that the model is missing sources of PM2.5 mass (e.g. nitrate aerosols in winter), which forces a compromise in which the constraint methodology rules in sulphate concentrations that are at the upper end of the uncertainty range to minimise the error for PM2.5. Although relaxing our constraint criteria offsets many effects of such structural errors, the shifting of these all-measurement constraint distributions away from the measurements indicates some structural error is still not fully accounted for. It is possible that our constraint would adjust better to account for this structural deficiency if we could directly specify a structural error term in the implausibility measure through $Var(S)$.





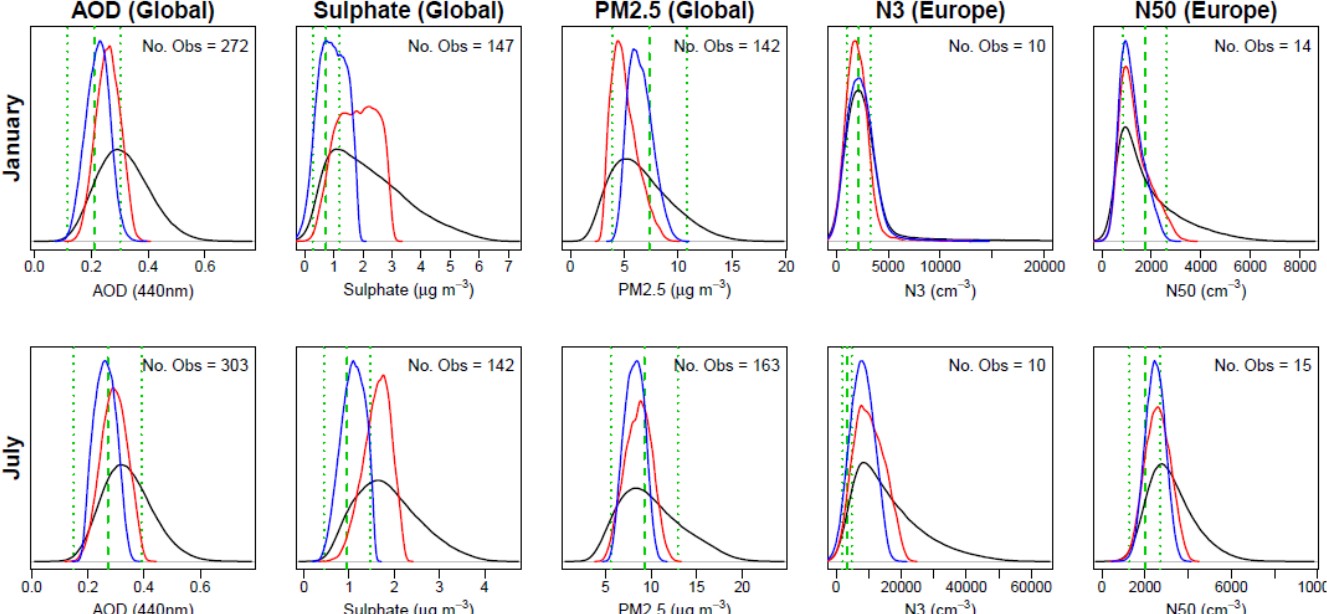

**Figure 8.** Comparison of the constrained model with the measurements for January (top) and July (bottom). The distributions were calculated as a mean over model grid boxes containing measurements. AOD, Sulphate and PM2.5 are global comparisons; $N_3$ and $N_{50}$ are Europe only comparisons due to the limited global coverage of these measurements in each month. The black line shows the prior (unconstrained) probability distribution of the model. The blue line shows the constrained model distribution when only measurements of each type are used in the constraint. The red line shows the constrained model distribution when all measurement types are used. The green dashed line shows the mean of the measurements and the dotted lines show the approximate 95% uncertainty range on an average observation that was accounted for in the constraint.

### 3.5 Unconstrained parameters

Several parameters are barely constrained or not constrained at all using all the measurements. Unconstrained microphysical processes or assumptions are the ageing rate of insoluble into soluble particles, the width of the lognormal accumulation mode, the hygroscopicity of organic material ($\kappa_{OC}$), the updraft speed and wet deposition rates. Among the emissions, unconstrained parameters are the emission rates of fossil fuel particles, degassing volcanic $SO_2$, DMS and dust emissions.

There are several potential reasons for the lack of constraint. It is possible that parts of the joint parameter space are ruled out, but with a negligible effect on the marginal parameter distribution (i.e., the ruled out parameter space is uniform across the parameter of interest). For example, wet deposition rates are directly compensated by emission rates and the ageing rate affects the wet removal rate. Another reason is that we did not include measurements in regions where the six variables are sensitive to these parameters. This is likely to be the case for DMS, volcanic and dust emissions given the relative lack of measurements over remote ocean regions and downwind of dust sources, which means these regions are not strongly weighted in the overall constraint process. Furthermore, some parameters may be more related to other aerosol properties that we have not used for constraint. For example, aging rates in the model are not constrained, likely because the ageing process predominantly affects the black carbon concentration which is not included as a measurement type in this study.



## 3.6 Implications for constraint of aerosol forcing

Figure 9 shows the nine most-important parameters for the uncertainty in global mean aerosol forcing in the PPE in terms of the forcing uncertainty they account for (Yoshioka et al., 2019). Some of these parameters are fairly strongly constrained by the measurements, but others are unconstrained. Within the joint parameter space of just these nine parameters there is considerable potential for model variants that compensate, thereby reducing the effectiveness of the constrained parameters on the forcing. It also needs to be borne in mind that global mean forcing is the sum of regional forcings, and in each region a different set of parameters is being constrained and may be constraining the same parameters to different parts of their range (Lee et al., 2016; Regayre et al., 2015).

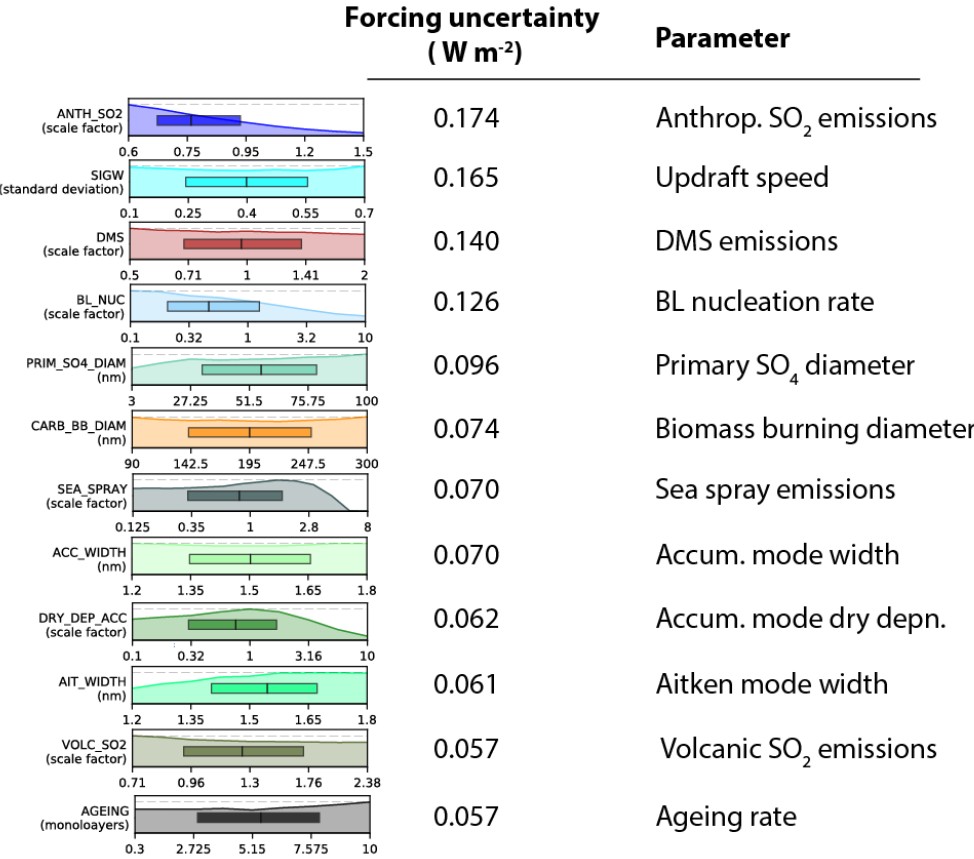

**Figure 9.** Ranked list of parameters that dominate the uncertainty in global mean aerosol radiative forcing in the ensemble (Yoshioka et al., 2019).

Figure 10 shows how the constrained parameters affect the uncertainty in predicted global annual mean net RF and its component parts due to aerosol-cloud interactions (RF$_{aci}$) and aerosol radiation interactions (RF$_{ari}$). (Note that this calculation of RF differs from that shown in Yoshioka et al. (2019), which used elicited parameter distributions when sampling over the





parameter uncertainty space, while we use uniform distributions for the sampling here). Table 3 shows the corresponding parameter distribution statistics (median, inter-quartile range, ±1σ range (on mean value) and 95% credible interval) for these forcing constraints.

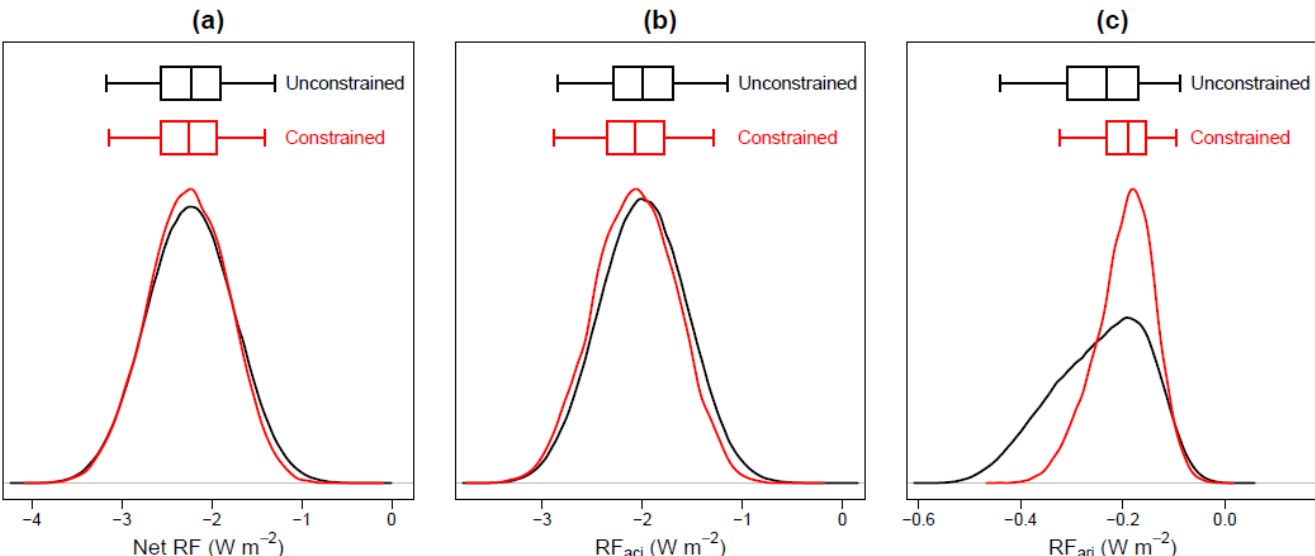

**Figure 10.** Effect of observational constraint using all measurement types on the probability distribution of global annual mean aerosol radiative forcing: (a) Net RF, (b) $RF_{aci}$ (aerosol-cloud interaction), and (c) $RF_{ari}$ (aerosol-radiation interaction). The black line shows the prior (unconstrained) distribution and the red line shows the constrained distribution. For each box and whisker plot, the box represents the interquartile range split at the median forcing (the line inside the box), and the whiskers extend to the lower and upper bounds of the 95% credible interval of the distribution.

| | Median | | Interquartile range | ±1σ Range | 95% Credible Interval | 95% CI Range Ratio (Constrained/Unconstrained) | 95% CI Reduction |
|---|---|---|---|---|---|---|---|
| **Net RF** | -2.23 | -2.26 | (-2.56 , -1.90) (-2.57 , -1.95) | (-2.71 , -1.75) (-2.71 , -1.81) | (-3.17 , -1.29) (-3.15 , -1.41) | 0.92 | 8% |
| **RF$_{aci}$** | -1.99 | -2.07 | (-2.29 , -1.69) (-2.36 , -1.78) | (-2.43 , -1.55) (-2.48 , -1.66) | (-2.84 , -1.15) (-2.88 , -1.28) | 0.94 | 6% |
| **RF$_{ari}$** | -0.23 | -0.19 | (-0.31 , -0.17) (-0.23 , -0.16) | (-0.33 , -0.15) (-0.26 , -0.14) | (-0.44 , -0.09) (-0.32 , -0.10) | 0.66 | 34% |

**Table 3.** Uncertainty distribution statistics (median, interquartile range, ±1σ range (on mean value) and 95% credible interval) for the global annual mean aerosol radiative forcing (Net RF, $RF_{aci}$ and $RF_{ari}$) from the unconstrained sample of 1 million model variants in black and the constrained sample of model variants from the all variables constraint in red. The final two columns show the ratio of the constrained to the unconstrained 95% credible interval and corresponding percentage reduction in this interval on constraint.

The net RF is dominated by $RF_{aci}$, which is only weakly constrained (Figure 10b) by 6%, in line with the net RF (Figure 10a).

This occurs because our constraint uses measurements of aerosol properties rather than cloud properties. Although the overall reduction in the $RF_{aci}$ uncertainty is weak, the pdfs in Figure 10b show a slight shift in $RF_{aci}$ to stronger forcings, with the median $RF_{aci}$ shifting from -1.99 W m$^{-2}$ in the prior (unconstrained) distribution to -2.07 W m$^{-2}$ after constraint. The likelihood



of the strength of RF$_{aci}$ being weaker than -1.5 W m$^{-2}$ (less negative) is reduced by 38% and the likelihood of it being stronger (more negative) than -2.5 W m$^{-2}$ is increased by 20%. In general, although the anthropogenic emissions were constrained to lower values (which should weaken the forcing), the sea spray emissions were constrained to lower values, which acts to strengthen the forcing (Carslaw et al., 2013a; Regayre et al., 2014). The 95% credible interval for the net RF is reduced by 8%.

The 95% credible interval of direct forcing, RF$_{ari}$, is reduced by 34% (Figure 10c) and the ±1σ range is reduced by 33%. RF$_{ari}$ is constrained most strongly by the PM2.5, AOD and sulphate measurements (not shown) but insignificantly by the OC, N$_3$ and N$_{50}$ measurements. The inconsistency between the constraints on PM2.5, AOD and sulphate (Figure 8) leads to inconsistency in the constraint on RF$_{ari}$. In particular, using just sulphate measurements results in an RF$_{ari}$ ±1σ range of -0.10 to -0.22 W m$^{-2}$, but using just PM2.5 measurements results in a range -0.20 to -0.36 W m$^{-2}$. This highlights the importance of detecting and fixing structural deficiencies in the model as well as the limitations of using single-variable emergent constraints.

It is important to note that the probability distribution of net aerosol RF includes values that, with current knowledge, would produce a net negative (greenhouse gas plus aerosol) forcing over the industrial period. The forcing is dominated by aerosol-cloud processes, which we have not attempted to constrain here. Nevertheless, the lack of constraint shows, as in Lee et al. (2016), a well-configured global aerosol model has little bearing on the uncertainty in RF$_{aci}$. In contrast, the constraint of RF$_{ari}$, which has not been attempted in our previous studies, is significant and encouraging.

## 4    Conclusions

We have used extensive point measurements of aerosol optical depth (AOD), PM2.5, sulphate mass, organic carbon mass, and the concentrations of particles larger than 50 nm dry diameter (N$_{50}$) and 3 nm (N$_3$) from surface sites, aircraft and ships to constrain uncertain aerosol parameters in a global aerosol-climate model. Twenty-six parameters related to aerosol emissions and processes were varied in a perturbed parameter ensemble and statistical emulators were used to generate a set of 1 million model variants that represent the model outputs for combinations of parameter values across the 26-dimensional uncertainty space. The plausibility of each model variant was tested against each measurement type in turn and then in combination using a history matching procedure based on an implausibility metric (Craig et al., 1996; Williamson et al., 2013). The resulting probability distributions of aerosol forcings can be considered as the "observationally plausible" ranges for the HadGEM3-UKCA model.

Observational constraint ruled out almost 98% of the 26-dimensional parameter space and the probability distributions of many parameter values were effectively constrained. Fourteen of the parameters were constrained to some extent, despite the fact that there are many ways in which parameter values can be combined to produce plausible results within the uncertainties. Constraint of a parameter means that the probability distribution of a parameter (and potentially its absolute range) is narrowed, and hence the likelihood of the parameter taking a particular range of values within its absolute range is increased; i.e., there





are more ways to combine these parameter values with values of the other 25 parameters to produce a plausible model. For two parameters, some of the individual prior elicited parameter ranges were ruled out entirely: very high sea spray emissions and very high residential carbonaceous emissions. The very highest BVOC emissions were nearly ruled out. However, for the remaining parameters it was not possible to entirely rule out any part of the prior range.

Parameter constraints are mostly, but not always, consistent across multiple measurement types, even though the different measurement types were made at very different locations and sometimes at different times of the year. For example, we often found consistent constraint of parameters related to the production of aerosol mass or number (emissions, nucleation, secondary aerosol mass) and parameters related to removal (condensation sink, deposition rates of gases and aerosols). There is also a very clear seasonal variation in the magnitude of constraint related to variations in the dominant processes.

The multivariate constraint has the following effect on the parameter probability distributions, which were assumed before constraint to be equally plausible between lower and upper bounds defined by expert elicitation:

1.  Boundary layer nucleation rates based on a sulphuric acid-organic mechanism (Metzger et al., 2010) are constrained to the low end of the elicited range, mainly from the $N_3$ measurements.

2.  The pH of cloud droplets, which controls aqueous-phase oxidation of $SO_2$ to form sulphate aerosol, is constrained to
be more likely in the middle of our elicited range but we were unable to rule out any of the pH values between 4.6 and 7.0. The constraint led to a reduction of 0.32 in the interquartile range to 5.24-6.12 (from 5.2-6.4 before constraint) and a reduction of 0.13 in the 95% credible interval to 4.69-6.84 (from 4.66-6.94 before constraint).

3.  Biomass burning emissions are weakly constrained by the PM2.5, AOD and OC measurements, with a reduced likelihood of the emissions exceeding a factor 2 above the default value. All lower emissions down to 0.25 times the
default value are equally likely as they were before constraint.

4.  Residential carbonaceous emissions are constrained primarily through a combination of PM2.5 and OC measurements. Emissions higher than about 2.7 times the default emission rate were effectively ruled out by the OC measurements as observationally implausible.

5.  The diameter of fossil fuel particles is constrained by $N_{50}$ measurements, with a reduced likelihood of being in the
range 30-45 nm, but diameters in the range 60-90 nm are unconstrained by any measurements.

6.  Sea spray emissions are constrained through a combination of AOD, PM2.5 and $N_{50}$ measurements. Emissions in the range 4.7-8.0 times the default value are ruled out but emissions in the range 0.125-2.8 times the default value are not strongly constrained by any of the measurements.

7.  Anthropogenic $SO_2$ emissions are strongly constrained to low values by AOD and sulphate measurements, with an
82% likelihood of being below the default value from the emission inventory.





8. BVOC emissions are constrained by AOD, PM2.5, OC, $N_{50}$ and $N_3$ measurements. The likelihood of either high or low emissions is reduced. On constraint, our median estimate corresponds to 86.5 Tg/y SOA production, with a 95% credible interval of 40-172 Tg/y. However, no values in the range 37-250 Tg/y are ruled out entirely.

9. The dry deposition rate of Aitken mode particles is weakly constrained to low values using OC and $N_3$ measurements and the dry deposition rate of accumulation mode particles is constrained to the middle of the range by AOD, sulphate, PM2.5, $N_{50}$ and $N_3$ measurements. The dry deposition rate of $SO_2$ is constrained to the upper part of the elicited range, with a 62% likelihood of it being above the default value.

Several parameters of importance to aerosol forcing were not well constrained, in particular parameters related to microphysical processes (primary sulphate particle diameter, the diameter of biomass burning particles, the width of the accumulation mode and the ageing rate). Dimethyl sulphide emissions were also not strongly constrained.

The prior (unconstrained) uncertainty (95% credible interval) in the pre-industrial to present-day net aerosol RF is reduced by 8%. The radiative forcing in the ensemble accounts for direct and indirect (cloud albedo) effects, but not cloud adjustments. $RF_{ari}$ uncertainty (95% credible interval) is reduced by 34%, but the net RF uncertainty is dominated by the $RF_{aci}$ uncertainty, which is reduced by only 6%. The recent assessment of aerosol forcing (Bellouin et al., 2019) adopted $\pm 1\sigma$ ranges to define the uncertainty. Our equivalent $\pm 1\sigma$ ranges are -0.14 to -0.26 W m$^{-2}$ for $RF_{ari}$ (reduction of 33% due to constraint) and -1.66 to -2.48 W m$^{-2}$ for $RF_{aci}$ (reduction of 7% due to constraint). The reduction in uncertainty is much larger for $RF_{ari}$ than $RF_{aci}$ because our constraints focus on aerosol properties rather than cloud properties.

Our results highlight the importance of using multiple measurement types to constrain aerosol-climate models. We have shown that use of a single measurement type, as is done in emergent constraint studies, would lead to an over-confident constraint. This is because potential structural deficiencies in our model prevent consistently good constraint across several measurement types. In particular, we showed that constraint using PM2.5 or sulphate aerosol measurements lead to probability distributions of $RF_{ari}$ that barely overlap. The final multivariate constraint on forcing is therefore a compromise that achieves reasonable agreement with all observations rather than being over-confidently constrained by one metric.

In terms of future directions and requirements to achieve better constraint, we make the following recommendations:

1. We need to understand and quantify model-measurement representativeness errors. The biggest challenge (and the factor that most limits the constraint, other than model structural error) is quantification of the representativeness error associated with comparing point measurements with a global model (Reddington et al., 2017; Schutgens et al., 2017, 2016a, 2016b). The ambiguity in deciding whether a model-measurement bias is related to structural error in the model or underestimation of the uncertainty terms was the main limiting factor in our constraint procedure. The representativeness uncertainties have been estimated based on model simulations at a few locations (Schutgens et al., 2016b), but they have not been measured, and we have no information about these uncertainties at other locations.



Small campaigns to characterise the space-time variability around existing sites and for potential new sites should be considered.

2. We need to understand and quantify structural deficiencies in the model. Our current approach, to assume zero structural error in our implausibility calculations, enables some structural errors in the model to be detected by comparing single and multi-measurement constraints. Ideally, to produce the most robust constraint possible, these structural errors should be accounted for directly using $Var(S)$ in the implausibility measure – or, better still, corrected in the model.

3. We should prioritise expansion of measurements to cover more long-term well-characterised sites. Such measurements will have much lower (or at least well-defined) representativeness error when used for model evaluation and constraint. Sites should be characterised in terms of how they represent a typical model grid box over a long period of time. These sites should be in diverse locations so that they help to constrain a wide range of model uncertainties (Reddington et al., 2017).

4. We should aim to dedicate part of field campaigns to routine, unbiased (or effectively random) sampling of aerosols across the scale of model grid boxes. Such measurements will also have much lower representativeness error than measurements that target specific processes or aerosol environments. Many field campaigns (particularly using aircraft) often prioritise measurements to explore aerosol processes, or to characterise particular aerosol environments (e.g., pollution plumes). Others, like the Atmospheric Tomography Mission emphasised the effective sampling of air masses in an unbiased way. A greater emphasis on such tomography missions, even just as part of a larger field campaign, would benefit model uncertainty reduction.

5. Additional aerosol measurements could be used to further constrain the parameters and forcing. We expect the following (already available) measurements would provide further constraint: i) black carbon measurements to help constrain the aerosol absorption component of radiative forcing, but also as a measure of aerosol removal rates; ii) $SO_2$ concentrations to avoid ambiguity between sulphate and PM2.5 constraints; iii) measurements from biomass burning regions to help constrain both the emissions and the size distribution of the particles; iv) vertical profiles of aerosol (Watson-Parris et al., 2019); v) more particle number size distribution information instead of just $N_3$ and $N_{50}$, which have not constrained the size distribution well enough.

6. Measured or derived process rates would be very useful because they would help to constrain model parameters directly, rather than relying on indirect constraint through measured state variables. For example, dry and wet deposition rates (Emerson et al., 2018) are required on the scale of model grid boxes (10's to 100's km). Similarly, direct estimates of particle formation and growth rates (Kerminen et al., 2018) would be useful, rather than just relying on integral particle number concentrations averaged over long periods.

It is very challenging to constrain model uncertainty using a large set of perturbed parameter model variants and extensive measurements of different types. However, the resulting ranges of model parameters and outputs (like radiative forcings)





estimated in this way are much more robust than those based on a very small number of models. Although our study is incomplete (not all parameters were perturbed and not all measurements were used) the outcome is an estimate of the "observationally plausible" range of aerosol forcings for the HadGEM3-UKCA aerosol-climate model. The ranges might be wider if we accounted for more sources of uncertainty and directly accounted for structural errors in the implausibility

5  calculations, but they could also be narrowed if we improved model structural deficiencies, reduced model-measurement-representativeness errors and used a wider set of measurement types.



## Appendix A

| Index | Parameter Name | Description |
|---|---|---|
| 1 | BL_Nuc | Boundary layer nucleation rate |
| 2 | Ageing | Ageing of hydrophobic aerosols (no of monolayers of soluble material) |
| 3 | Acc_Width | Modal width of accumulation modes (nm) |
| 4 | Ait_Width | Modal width of Aitken modes (nm) |
| 5 | Cloud_pH | pH of cloud droplets (used to calculate the conversion of $SO_2$ into sulphate) |
| 6 | Carb_FF_Ems | Carbonaceous fossil fuel emissions scale factor |
| 7 | Carb_BB_Ems | Carbonaceous biomass burning emissions scale factor |
| 8 | Carb_Res_Ems | Carbonaceous residential (biofuel) emissions scale factor |
| 9 | Carb_FF_Diam | Carbonaceous fossil fuel emission diameter (nm) |
| 10 | Carb_BB_Diam | Carbonaceous biomass burning emission diameter (nm) |
| 11 | Carb_Res_Diam | Carbonaceous residential (biofuel) emission diameter (nm) |
| 12 | Prim_SO4_Frac | Mass fraction of $SO_2$ converted to new sulphate particles in power plant plumes |
| 13 | Prim_SO4_Diam | Mode diameter of new sub-grid sulphate particles (nm) |
| 14 | Sea_Spray | Sea spray aerosol scale factor |
| 15 | Anth_SO2 | Anthropogenic $SO_2$ emission scale factor |
| 16 | Volc_SO2 | Volcanic $SO_2$ emission scale factor |
| 17 | BVOC_SOA | Biogenic secondary aerosol formation from volatile organic compounds scale factor |
| 18 | DMS | Dimethyl sulphide surface ocean concentration scale factor |
| 19 | Dry_Dep_Ait | Aitken mode dry deposition velocity scale factor |
| 20 | Dry_Dep_Acc | Accumulation mode dry deposition velocity scale factor |
| 21 | Dry_Dep_SO2 | $SO_2$ dry deposition velocity scale factor |
| 22 | Kappa_OC | Kappa-Kohler coefficient of organic carbon |
| 23 | Sig_W | Updraft vertical velocity standard deviation (used to calculate the activation of aerosols into cloud drops) |
| 24 | Dust | Dust emission scale factor |
| 25 | Rain_Frac | Fraction of cloud-covered area in large-scale clouds where aerosol scavenging by rain drops occurs |
| 26 | Cloud_Ice_Thresh | Threshold of cloud ice fraction above which nucleation scavenging is suppressed (restricting further activation of aerosols into cloud drops) |

**Table A1.** The 26 aerosol parameters included in the AER PPE. Further details are provided in a separate publication (Yoshioka et al., 2019).



| AOD | | All-months constraint choices (LHS Fig. 5) | | | Joint all-variables-months constraint choices (RHS Fig 5) | | |
|---|---|---|---|---|---|---|---|
| Month | No. Obs used in constraint | Threshold Implausibility | Tolerance (No. Obs) | % Variants Retain | Threshold Implausibility | Tolerance (No. Obs) | % Variants Retain |
| Jan | 272 / 294 | 3.5 | 33 | 40.03 | 4 | 76 | 75.60 |
| Feb | 274 / 301 | 3.5 | 27 | 42.37 | 3.5 | 82 | 75.71 |
| Mar | 284 / 309 | 3.5 | 23 | 40.54 | 3.5 | 80 | 75.78 |
| Apr | 289 / 316 | 3.5 | 17 | 39.70 | 3.5 | 52 | 76.47 |
| May | 292 / 322 | 3.5 | 12 | 34.55 | 3.5 | 47 | 76.17 |
| Jun | 295 / 320 | 3.5 | 24 | 43.10 | 3.5 | 53 | 74.13 |
| Jul | 303 / 323 | 3.5 | 24 | 43.38 | 3.5 | 48 | 75.46 |
| Aug | 310 / 326 | 3.5 | 19 | 35.51 | 3.5 | 50 | 74.06 |
| Sep | 303 / 321 | 3.5 | 24 | 43.09 | 3.5 | 61 | 74.65 |
| Oct | 285 / 315 | 3.5 | 23 | 39.57 | 3.5 | 80 | 75.97 |
| Nov | 273 / 309 | 3.5 | 27 | 41.91 | 3.5 | 93 | 74.75 |
| Dec | 267 / 298 | 3.5 | 43 | 41.44 | 4 | 91 | 75.82 |
| ALL | 3447 / 3754 | | | 19.01 | | | 60.42 |
| Sulphate | | All-months constraint choices (LHS Fig. 5) | | | Joint all-variables-months constraint choices (RHS Fig 5) | | |
| Month | No. Obs used in constraint | Threshold Implausibility | Tolerance (No. Obs) | % Variants Retain | Threshold Implausibility | Tolerance (No. Obs) | % Variants Retain |
| Jan | 147 / 149 | 3.5 | 3 | 38.76 | 4 | 50 | 75.42 |
| Feb | 164 / 148 | 3.5 | 3 | 38.79 | 4 | 38 | 74.64 |
| Mar | 144 / 151 | 3.5 | 6 | 42.70 | 4 | 46 | 74.99 |
| Apr | 149 / 151 | 3.5 | 6 | 38.66 | 4 | 51 | 75.22 |
| May | 146 / 149 | 3.5 | 9 | 41.38 | 4.5 | 35 | 74.60 |
| Jun | 142 / 150 | 3.5 | 6 | 38.19 | 3.5 | 48 | 74.40 |
| Jul | 142 / 148 | 3.5 | 9 | 41.75 | 4 | 40 | 75.19 |
| Aug | 143 / 148 | 3.5 | 9 | 38.97 | 4 | 46 | 75.00 |
| Sep | 144 / 147 | 3.5 | 3 | 39.13 | 4 | 46 | 74.77 |
| Oct | 143 / 147 | 3.5 | 3 | 38.03 | 4 | 46 | 75.02 |
| Nov | 143 / 146 | 3.5 | 3 | 40.21 | 4 | 46 | 75.52 |
| Dec | 144 / 147 | 3.5 | 3 | 41.52 | 4 | 49 | 72.42 |
| ALL | 1733 / 1781 | | | 27.73 | | | 64.90 |
| PM2.5 | | All-months constraint choices (LHS Fig. 5) | | | Joint all-variables-months constraint choices (RHS Fig 5) | | |
| Month | No. Obs used in constraint | Threshold Implausibility | Tolerance (No. Obs) | % Variants Retain | Threshold Implausibility | Tolerance (No. Obs) | % Variants Retain |
| Jan | 142 / 168 | 3.5 | 20 | 41.52 | 3.5 | 40 | 74.02 |
| Feb | 145 / 168 | 3.5 | 20 | 36.27 | 3.5 | 38 | 73.51 |



| Month | No. Obs used in constraint | Threshold Implausibility | Tolerance (No. Obs) | % Variants Retain | Threshold Implausibility | Tolerance (No. Obs) | % Variants Retain |
|---|---|---|---|---|---|---|---|
| **Mar** | 154 / 170 | 3.5 | 15 | 39.14 | 3.5 | 34 | 75.09 |
| **Apr** | 152 / 170 | 3.5 | 9 | 43.70 | 3.5 | 24 | 72.93 |
| **May** | 154 / 167 | 3.5 | 9 | 43.16 | 3.5 | 22 | 75.83 |
| **Jun** | 158 / 170 | 3.5 | 13 | 34.89 | 3.5 | 32 | 74.91 |
| **Jul** | 163 / 172 | 3.5 | 16 | 40.27 | 3.5 | 36 | 74.08 |
| **Aug** | 161 / 169 | 3.5 | 13 | 36.70 | 3.5 | 39 | 75.68 |
| **Sep** | 154 / 166 | 3.5 | 9 | 42.71 | 3.5 | 31 | 75.37 |
| **Oct** | 151 / 165 | 3.5 | 12 | 45.71 | 3.5 | 24 | 75.59 |
| **Nov** | 142 / 168 | 3.5 | 14 | 38.76 | 3.5 | 31 | 73.69 |
| **Dec** | 148 / 169 | 3.5 | 24 | 40.51 | 3.5 | 44 | 75.52 |
| **ALL** | 1824 / 2022 | | | 10.80 | | | 48.89 |

| OC | | **All-months constraint choices (LHS Fig. 5)** | | | **Joint all-variables-months constraint choices (RHS Fig 5)** | | |
|---|---|---|---|---|---|---|---|
| **Month** | No. Obs used in constraint | Threshold Implausibility | Tolerance (No. Obs) | % Variants Retain | Threshold Implausibility | Tolerance (No. Obs) | % Variants Retain |
| **Jan** | 3 / 6 | 3.5 | 0 | 63.60 | 4.5 | 0 | 76.25 |
| **Feb** | 10 / 14 | 3.5 | 3 | 58.59 | 3.5 | 4 | 86.56 |
| **Mar** | 41 / 82 | 3.5 | 2 | 57.69 | 3.5 | 3 | 72.06 |
| **Apr** | 49 / 74 | 3.5 | 2 | 68.29 | 3.5 | 3 | 76.20 |
| **May** | 20 / 23 | 3.5 | 1 | 56.91 | 3.5 | 2 | 76.69 |
| **Jun** | 22 / 23 | 3.5 | 3 | 71.67 | 3.5 | 4 | 76.64 |
| **Jul** | 18 / 23 | 3.5 | 2 | 57.40 | 3.5 | 4 | 73.36 |
| **Aug** | 21 / 23 | 3.5 | 2 | 65.05 | 3.5 | 3 | 72.19 |
| **Sep** | 17 / 22 | 4 | 4 | 67.45 | 3.5 | 5 | 77.65 |
| **Oct** | 24 / 41 | 3.5 | 4 | 72.35 | 3.5 | 4 | 72.35 |
| **Nov** | 15 / 37 | 4.5 | 0 | 58.57 | 3.5 | 1 | 81.81 |
| **Dec** | 11 / 15 | 3.5 | 1 | 71.62 | 3.5 | 1 | 71.62 |
| **ALL** | 251 / 383 | | | 6.94 | | | 20.00 |

| $N_3$ | | **All-months constraint choices (LHS Fig. 5)** | | | **Joint all-variables-months constraint choices (RHS Fig 5)** | | |
|---|---|---|---|---|---|---|---|
| **Month** | No. Obs used in constraint | Threshold Implausibility | Tolerance (No. Obs) | % Variants Retain | Threshold Implausibility | Tolerance (No. Obs) | % Variants Retain |
| **Jan** | 12 / 13 | 3.5 | 0 | 95.74 | 3.5 | 0 | 95.74 |
| **Feb** | 12 / 13 | 3.5 | 0 | 96.60 | 3.5 | 0 | 96.60 |
| **Mar** | 12 / 13 | 3.5 | 0 | 85.80 | 3.5 | 0 | 85.80 |
| **Apr** | 11 / 12 | 3.5 | 0 | 68.78 | 3.5 | 1 | 75.35 |
| **May** | 11 / 12 | 3.5 | 0 | 56.05 | 3.5 | 4 | 77.67 |
| **Jun** | 12 / 12 | 3.5 | 1 | 58.61 | 3.5 | 4 | 75.70 |
| **Jul** | 13 / 13 | 3.5 | 1 | 57.15 | 3.5 | 4 | 73.41 |
| **Aug** | 13 / 13 | 3.5 | 1 | 53.12 | 3.5 | 4 | 73.59 |
| **Sep** | 13 / 13 | 3.5 | 0 | 67.29 | 3.5 | 1 | 74.97 |

| Month | No. Obs used in constraint | Threshold Implausibility | Tolerance (No. Obs) | % Variants Retain | Threshold Implausibility | Tolerance (No. Obs) | % Variants Retain |
|---|---|---|---|---|---|---|---|
| Oct | 12 / 13 | 3.5 | 0 | 76.94 | 3.5 | 0 | 76.94 |
| Nov | 12 / 13 | 3.5 | 1 | 97.44 | 3.5 | 1 | 97.44 |
| Dec | 10 / 12 | 3.5 | 0 | 94.03 | 3.5 | 0 | 94.03 |
| ALL | 143 / 152 | | | 44.03 | | | 61.58 |
| $N_{50}$ | | **All-months constraint choices (LHS Fig. 5)** | | | **Joint all-variables-months constraint choices (RHS Fig 5)** | | |
| **Month** | **No. Obs used in constraint** | **Threshold Implausibility** | **Tolerance (No. Obs)** | **% Variants Retain** | **Threshold Implausibility** | **Tolerance (No. Obs)** | **% Variants Retain** |
| **Jan** | 74 / 77 | 3.5 | 3 | 55.41 | 3.5 | 9 | 76.09 |
| **Feb** | 82 / 90 | 3.5 | 7 | 53.78 | 3.5 | 13 | 75.28 |
| **Mar** | 132 / 148 | 3.5 | 8 | 50.40 | 3.5 | 16 | 74.89 |
| **Apr** | 177 / 199 | 3.5 | 25 | 52.59 | 3.5 | 32 | 71.65 |
| **May** | 62 / 64 | 3.5 | 2 | 53.39 | 3.5 | 7 | 74.25 |
| **Jun** | 88 / 96 | 3.5 | 4 | 48.43 | 3.5 | 9 | 73.21 |
| **Jul** | 102 / 115 | 3.5 | 4 | 48.30 | 3.5 | 10 | 74.10 |
| **Aug** | 97 / 109 | 3.5 | 4 | 58.29 | 3.5 | 8 | 73.51 |
| **Sep** | 114 / 133 | 3.5 | 7 | 52.34 | 3.5 | 14 | 76.95 |
| **Oct** | 112 / 119 | 3.5 | 9 | 45.93 | 3.5 | 16 | 72.77 |
| **Nov** | 113 / 155 | 3.5 | 20 | 49.80 | 3.5 | 27 | 77.18 |
| **Dec** | 62 / 67 | 3.5 | 9 | 46.17 | 3.5 | 14 | 77.70 |
| **ALL** | 1215 / 1372 | | | 9.99 | | | 40.45 |
| **Final % variants retain for the joint all-variables-months constraint:** | | | | | | | **2.085** |

**Table A2.** The choices of the number of measurements (tolerance, $T$) allowed to exceed a threshold implausibility ($\theta$) for constraint on each of the monthly observed aerosol properties, along with the corresponding percentage of the large sample of 1 million model variants (that covers the PPE parametric uncertainty) that is retained on constraint for each choice. The choices on the left correspond to those used for the all-months constraints shown on the left of Figure 6. The choices on the right correspond to those used for the joint all-variables-months constraint shown on the right in Figure 6. The percentage of variants retained is also shown for the combined all-months constraints for each variable (ALL) as well as for the joint all-variables-months constraint (final row).

## Appendix B: Description of the data sets used in this study

For the AERONET Aerosol Optical Depth (AOD) data used in this study, we thank the PIs of the AERONET sites used for maintaining their instruments and providing their data to the community. We also acknowledge AERONET for their continuous efforts in providing high-quality measurements and derivative products. All data used in this work can be accessed through the AERONET web page: http://aeronet.gsfc.nasa.gov/.

For the sulphate data used in this study, we acknowledge the EMEP (http://ebas.nilu.no/; Tørseth et al., 2012), IMPROVE (http://views.cira.colostate.edu/fed/) and EANET (http://www.eanet.asia/product/index.html) measurement networks for making their measurement data available, along with all data managers involved in data collection. Additional ground station observations from the SORPES (Station for Observing Regional Processes of the Earth System) monitoring station in Nanjing, China (Ding et al., 2016) are also included. Data on the Acid Deposition in the East Asian Region was provided





from the Network Center for EANET, https://monitoring.eanet.asia/document/public/index (last accessed: 7 June 2018).
IMPROVE is a collaborative association of state, tribal, and federal agencies, and international partners. US Environmental
Protection Agency is the primary funding source, with contracting and research support from the National Park Service. The
Air Quality Group at the University of California, Davis is the central analytical laboratory, with ion analysis provided by
Research Triangle Institute, and carbon analysis provided by Desert Research Institute.

For the PM2.5 data used in this study, we acknowledge the IMPROVE (http://views.cira.colostate.edu/fed/), WMO GAW–
WDCA (https://www.gaw-wdca.org; http://ebas.nilu.no/; Tørseth et al., 2012), A-PAD (Atanacio et al., 2016), and NAPS
(Galarneau et al., 2016; http://maps-cartes.ec.gc.ca/rnspa-naps/data.aspx?lang=en) measurement networks for making their
measurement data available, along with all data managers involved in data collection. Further ground station measurements
are included from sites in Australia (ANSTO stations: Cohen and Atanacio, 2015), South America (Artaxo et al., 2013),
Taiwan (Fang and Chang, 2010), South Africa (Vakkari et al., 2013) and Nanjing, China (SORPES station; Ding et al.,
2016). The PM2.5 data for Europe was obtained from the world data centre for aerosol (WDCA), and we thank the following
data providers to this network: Adamos Adamides (Cyprus), Jacobus P. J. Berkhout (Netherlands), Elke Bieber (Germany),
Tanja Bolte (Slovenia), Geoff Broughton (United Kingdom), Darius Ceburnis (Ireland), Anna Degorska (Poland), Iveta
Dubakova (Latvia),Fermin Elizaga (Spain), Marina Froehlich (Austria), Marina Frolova (Latvia), Robert Gehrig
(Switzerland), Alberto Gonzalez (Spain), C. Gruening (Italy), Savvas Kleanthous (Cyprus), Manuel Lambas (Spain), Maj
Britt Larka Abellan (Spain), Marijana Murovec (Slovenia), Jaroslav Pekarek (Czech Republic), Noemi Perez (Spain), Cinzia
Perrino (Italy), Jean-Philippe Putaud (Italy), Xavier Querol (Spain), Stephane Sauvage (France), Karin Sjoberg (Sweden),
Andre Sonntag (Germany), Gerald Spindler (Germany), D. P. J. Swart (Netherlands), Karin Uhse (Germany), Milan Vana
(Czech Republic), Keith Vincent (UK), P. Aalto (Finland), M. Kulmala (Finland), Anne-Gunn Hjellbrekk, (Norway) and
Aas Wenche (Norway).

The N3 data used in this study was obtained from the EBAS ACTRIS database (Asmi et al., 2013; https://www.actris.eu/;
http://ebas.nilu.no/), collated via the Global Aerosol Synthesis and Science Project, GASSP (Reddington et al., 2017) and
public data on the EBAS database.  The EBAS database has largely been funded by the UNECE CLRTAP (EMEP) and
AMAP and through NILU internal resources. Specific developments have been possible due to projects like EUSAAR (EU-
FP5; EBAS web interface), EBAS Online (Norwegian Research Council INFRA; upgrading the database platform) and
HTAP (European Commission DG-ENV; import and export routines to build a secondary repository in support of
http://www.htap.org; last access: 4 April 2019). A large number of specific projects have supported development of data and
meta data reporting schemes in dialog with data providers (EU; CREATE, ACTRIS and others). Through ACTRIS, the
research leading to the these results has received funding from the European Union's Horizon 2020 research and innovation
programme under grant agreement No 654109.  For a complete list of programmes and projects for which EBAS serves as a
database, please consult the information box in the Framework filter of the web interface. These are all highly acknowledged
for their support.



The observations of OC used in this study are collated from 7 observational campaigns and supplemented by additional ground station observations. The campaign data were collated via GASSP and are derived from size distribution measurements taken during the following campaigns: VOCALS (NERC Grant: NE/F019874/1; Allen et al., 2011; Hawkins et al., 2010; Wood et al., 2011), CalNex (Ryerson et al., 2013), WACS (Quinn et al., 2014), ICEALOT (Frossard et al., 2011), DYNAMO (DeWitt et al., 2013), NEAQS-2004 (Quinn et al., 2006; Wang et al., 2007), TEXAQS06 (Bates et al., 2008), RHaMBLe (NERC Grants NE/D006570/1, NE/E011454/1; Allan et al., 2009) and ACCACIA (NERC Grant NE/I028696/1; Allan et al., 2015). The OC ground station observations used are from the AMS Global Database which has a worldwide coverage (Zhang et al.), along with data for further European sites from ACTRIS (https://www.actris.eu/; http://ebas.nilu.no/; collated via GASSP) and data from individual stations including Chilbolton (England; Crippa et al., 2014), COPS (NERC Grant NE/E016200/1; Hornisgrinde, Germany; Irwin et al., 2010; Jones et al., 2011), Holme Moss (England; Liu et al., 2011), OP3 (NERC Grant NE/D004624/1; South-East Asia; Hewitt et al., 2010) and MC4 (NERC Grant NE/H008136/1; Weybourne, England; Liu et al., 2013) collated via GASSP, the SORPES site in Nanjing, China (Ding et al., 2016), and AMF stations in the USA and north-east Atlantic (Atmospheric Radiation Measurement (ARM) user facility, 2014a, 2014b). The OC data at the AMF stations (USA and north-east Atlantic) were obtained from the Atmospheric Radiation Measurement (ARM) Program sponsored by the U.S. Department of Energy, Office of Science, Office of Biological and Environmental Research, Climate and Environmental Sciences Division. The AMS Global Database ground station data covers the following sites: Barcelona (Minguillón et al., 2011; Mohr et al., 2012), Beijing (Sun et al., 2010), Blodgett Forest (Farmer et al., 2011), Boulder (Nemitz et al., 2008), Cape Hedo, Chebogue Point, Cheju Island (Topping et al., 2004), Chilbolton, Cool (Setyan et al., 2012), Duke Forest (Stroud et al., 2007), Edinburgh (Allan et al., 2003a), Finokalia (Hildebrandt et al., 2010), Fukue Island (Takami et al., 2005), Helsinki (Timonen et al., 2013), Houston (Canagaratna et al., 2007), Hyytiala (Allan et al., 2006), Jungfraujoch (Ng et al., 2010), Komaba (Takegawa et al., 2006), K-Puszta, Mace Head (Dall'Osto et al., 2010), Mainz (Ng et al., 2010), Manaus (Chen et al., 2009), Manchester (Allan et al., 2003b), Melpitz (Poulain et al., 2011), Mexico City (Aiken et al., 2009), Montseny, New York City (Drewnick et al., 2004; Sun et al., 2011; Weimer et al., 2006), Pasadena (Hayes et al., 2013), Pinnacle State Park (Bae et al., 2007), Pittsburgh (Zhang et al., 2005), Point Reyes National Seashore | ARM Mobile Facility (AMF) (Ervens et al., 2010), Puy de Dome (Freney et al., 2011), Riverside | SOAR field site (Docherty et al., 2011; Williams et al., 2010), San Pietro Capofiume, Storm Peak, Trinidad Head (Millet et al., 2004), Vancouver (Alfarra et al., 2004; Boudries et al., 2004), Weybourne Atmospheric Observatory, Whistler Mountain (Sun et al., 2009), Whiteface Mountain (Hogrefe et al., 2004), and Writtle Agricultural College.

The observations of $N_{50}$ used in this study are collated from 19 observational campaigns and supplemented by additional ground station observations. The campaign data, collated via GASSP, and derived from size distribution measurements taken during the following campaigns: ACE1 (Bates et al., 1998; Clarke et al., 1998), VOCALS (NERC Grant: NE/F019874/1; Allen et al., 2011; Hawkins et al., 2010; Wood et al., 2011), DOE ARM MAGIC (Lewis and Teixeira, 2015), CalNex (Ryerson et al., 2013), WACS (Quinn et al., 2014), NEAQS-2002 (Bates et al., 2005; Quinn and Bates, 2005), ARCTAS (McNaughton et

al., 2011), ASCOS (Heintzenberg and Leck, 2012; Tjernström et al., 2014), ICEALOT (Frossard et al., 2011), AEROSOL99 (Bates et al., 2001), DYNAMO (DeWitt et al., 2013), INDOEX (Quinn and Bates, 2005; Ramanathan et al., 2001), PEM-Tropics-A (Fenn et al., 1999), PEM-Tropics- B (Raper et al., 2001), PASE (Hudson and Noble, 2009), NAURU99 (Long and McFarlane, 2012), ACE-ASIA (Bates et al., 2004; Huebert et al., 2003), NEAQS-2004 (Quinn et al., 2006; Wang et al., 2007)

and TEXAQS06 (Bates et al., 2008). The $N_{50}$ ground station observations used, collated via GASSP and public data on the EBAS database, are from Canada (Jeong et al., 2010; Leaitch et al., 2013; Takahama et al., 2011), South Africa (Vakkari et al., 2013), the Russian Arctic (Asmi et al., 2016), India (Hyvärinen et al., 2010), Antarctica (Fiebig et al., 2009) and European sites (Asmi et al., 2011). ASCOS (the Arctic Summer Cloud Ocean Study) was funded by the Knut and Alice Wallenberg Foundation and DAMOCLES (EU 6th Framework Program). The Swedish Polar Research Secretariat provided access to the

icebreaker Oden and logistical support.

**Data Availability**

Measurement data can be sourced from the databases and data providers as outlined in Appendix B and Table S1 in the supplementary data file. Model data and analysis code can be made available upon request from the corresponding author. The authors welcome use of the perturbed parameter ensemble for advancing climate research.

**Author Contributions**

JSJ applied the statistical methodology and generated and analysed the results. JSJ, LAR and KSC wrote the article. JSJ, LAR, KSC, KJP, STT, JB, DMHS, JWR and NAJS contributed to the analysis and interpretation of results. KJP, MY, LAR, NAJS, DGP, KSC and JSJ helped prepare the model configuration that served as the template for the PPE. LAR and JSJ designed the experiments. All PPE simulations were created by MY. LAR and JSJ elicited probability density functions of all aerosol

parameters, and KSC, KJP, MY, and STT participated (alongside many other experts) in the formal elicitation process. DL, JDA, HC, AD, DDC, AA, VV and EA contributed measurement data to the study from individual measurement campaigns.

**Competing Interests**

The authors declare that they have no conflict of interest.

**Acknowledgements**

This research was funded by the Natural Environment Research Council (NERC) under Grants NE/J024252/1 (GASSP), NE/I020059/1 (ACID-PRUF) and NE/P013406/1 (A-CURE); the European Union ACTRIS-2 project under grant 262254; the National Centre for Atmospheric Science (Yoshioka, Carslaw); and by the UK–China Research and Innovation Partnership



Fund through the Met Office Climate Science for Service Partnership (CSSP) China as part of the Newton Fund. We made use of the N8 HPC facility funded from the N8 consortium and an Engineering and Physical Sciences Research Council Grant to use ARCHER (EP/K000225/1) and the JASMIN facility (www.jasmin.ac.uk/) via the Centre for Environmental Data Analysis funded by NERC and the UK Space Agency and delivered by the Science and Technology Facilities Council. We acknowledge the following additional funding: the Royal Society Wolfson Merit Award (Carslaw); a doctoral training grant from NERC and a CASE studentship with the Met Office Hadley Centre (Regayre); the H2020 project INTAROS: Integrated Arctic Observing System, project ID: 727890 (Asmi); NERC grants to the University of Manchester: NE/F019874/1 (VOCALS); NE/D004624/1 (OP3); NE/H008136/1 (MC4); NE/D006570/1, NE/E011454/1 (RHaMBLe); NE/E016200/1 (COPS); NE/I028696/1 (ACCACIA).

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
