# Peer review of "Robust observational constraint of uncertain aerosol processes and emissions in a climate model and the effect on aerosol radiative forcing"

_Atmospheric Chemistry and Physics, 2019_

## Referee Comment (RC1) · Anonymous Referee #1 · 4 Dec 2019

This manuscript investigates the constraint on aerosol forcing uncertainties by observations of aerosol mass, number, and optical depth. The manuscript builds upon earlier work from the authors using statistical fits of climate model simulations in order to fully sample the uncertainty space of many (27) uncertain aerosol parameters in the model. The authors use the observations to determine which portions of the parameter space provide poor agreement with the observations and can be ruled out. They then determine how much this narrows the direct and cloud-albedo indirect forcing uncertainty ranges.

I feel this paper is ready for publication in ACP once several minor comments have

been addressed.

P6 L5: Have the authors thought about how much additional information would be added if they had the simulations/emulators built over all of the years with observations such that observations could compared with the same month-year? What if the exact days/times could be compared? Certainly each of these steps requires a jump in computational investment, so it doesn't make sense to add here. But if the authors have ideas about how much it could help, it would be an interesting thing to discuss (perhaps in the conclusions?).

P6: At least some of the PM2.5 networks where sulfate concentrations were taken also have OC or OA. Why was OA from those sites not used (e.g. IMPROVE)?

P12 L11-12: The incorrect model year relative the measurement year could also cause this mismatch, right? I could imagine this being a big issue for regions with significant biomass burning emissions, which have significant interannual variability.

Figure 6: I appreciate the challenge in creating this figure. Is there any way that it can be made more crisp. The combination of very small fonts and rough resolution makes it very hard to read (at least when printed).

P18 L31: Need to say "northern-hemisphere winter" here (it's said in the next sentence), but it should be clear when "winter" is first used.

P19 L13: Is nitrate not included at all in the simulations? It seems like that could throw things off a lot (which is I guess what you're saying). It would be good to make it clear that nitrate was not included (if it's not) or if it's just incorrect.

Have the authors looked at how much the observations impact the direct aerosol *effect* (the radiative effect of natural and anthro aerosols relative to no aerosols) rather than the radiative forcing? I wonder how different the level of constraint would be. This isn't necessary for the paper unless the authors can do this quickly and agree that it's interesting.

---

## Referee Comment (RC2) · Anonymous Referee #2 · 12 May 2020

This manuscript investigates the constraint on aerosol forcing uncertainties by observations of AOD, PM2.5, particle number concentrations, sulphate and organic mass concentrations. The manuscript is scientifically sound, and of a high scientific quality. In my opinion, the manuscript is ready for publication in ACP, since it presents not significant shortcomings in the scientific significance or quality.

---

## Author Comment (AC1) · 22 Jun 2020

In our response, reviewer comments are marked in bold, our responses and original text in plain text, and altered text in the paper in bold italic.

**Response to reviewer 1 (Anonymous reviewer)**

**This manuscript investigates the constraint on aerosol forcing uncertainties by observations of aerosol mass, number, and optical depth. The manuscript builds upon earlier work from the authors using statistical fits of climate model simulations in order to fully sample the uncertainty space of many (27) uncertain aerosol parameters in the model. The authors use the observations to determine which portions of the parameter space provide poor agreement with the observations and can be ruled out. They then determine how much this narrows the direct and cloud-albedo indirect forcing uncertainty ranges.**

**I feel this paper is ready for publication in ACP once several minor comments have been addressed.**

We thank the reviewer for their interesting and useful comments on our manuscript, and respond in full to the comments below. We would like to clarify here that the PPE used in this study perturbed only 26 aerosol parameters. We did simultaneously create a 27 parameter PPE alongside this PPE (see Yoshioka et al., 2019), but that ensemble has not been analysed here.

**Comment 1:**

**P6 L5: Have the authors thought about how much additional information would be added if they had the simulations/emulators built over all of the years with observations such that observations could compared with the same month-year? What if the exact days/times could be compared? Certainly each of these steps requires a jump in computational investment, so it doesn't make sense to add here. But if the authors have ideas about how much it could help, it would be an interesting thing to discuss (perhaps in the conclusions?).**

This is an interesting point and we have considered this. Accounting for all sources of model-measurement uncertainty is important, and our strategy here is to account for as many sources as possible in our definition of the implausibility metric. The representativeness error terms in the metric would be smaller if we had model output for several years spanning the measurement data, and at a higher temporal and spatial resolution, which would give us more confidence in ruling out implausible model variants.

We have found that inter-annual variability can account for around 30 to 90% of the model-measurement comparison representation uncertainty. Hence, making direct year-on-year comparisons between all measurements and the PPE could significantly reduce the effect of inter-annual variability on the comparison process. However, the large set of observations used in this study contains campaign data from many different years, and so this strategy is not computationally feasible here, as the reviewer remarks.

We have included some additional discussion about this issue in the conclusions section of our manuscript (section 5), in point 1 of our list of future recommendations for future directions and requirements to achieve better constraint. The added text is:

*'Alternatively, representation errors could be reduced by increasing the temporal and spatial resolution of our model, and the effect of inter-annual variability could be reduced if we always*

*compare the model to measurements for the correct year. In our calculations of model variant implausibility, inter-annual variability accounts for around 30 to 90% of the model-measurement comparison representation uncertainty. With smaller representation errors in the model-measurement comparison, we could more confidently reject implausible model variants. However, this approach is not currently computationally feasible.*'

We appreciate the reviewer pointing out the importance of reducing the causes of model-measurement uncertainty and considering these when designing experiments. We are currently creating a PPE (on the NERC-funded A-CURE project) with output at a much higher temporal resolution that spans the period covered by multiple important flight campaigns.

**Comment 2:**

**P6: At least some of the PM2.5 networks where sulfate concentrations were taken also have OC or OA. Why was OA from those sites not used (e.g. IMPROVE)?**

For this study we have used all OC/OA measurement data that had been processed into a 'model-ready' format for comparison through the NERC GASSP project, and stored in the GASSP database, at the time of analysis. Unfortunately, this did not include OA data from some of the measurement networks like IMPROVE. More measurements are being added to the GASSP database and processed to a format for model comparison, and we anticipate that such data will be available for future work. In the next phases of our model-measurement comparison work, we will endeavour to ensure that more OA data from networks such as IMPROVE are incorporated to further strengthen our resulting comparisons with this variable.

**Comment 3:**

**P12 L11-12: The incorrect model year relative the measurement year could also cause this mismatch, right? I could imagine this being a big issue for regions with significant biomass burning emissions, which have significant interannual variability.**

Yes, the inter-annual variability could also potentially cause such a mismatch. This text in Section 2.4.3 has been adjusted to also include this possibility. It now reads:

'…We assume that this large implausibility for the significant majority of variants indicates either there is a structural error in the model or that the model is unable to represent these point measurements because of its low spatial and temporal resolution*. An alternative explanation is a mismatch in the model's meteorological year to the year of the measurement* (section 2.4.1). We flag these measurements for further investigation of potential structural errors or underestimated error terms (these are not examined further in this study).'

**Comment 4:**

**Figure 6: I appreciate the challenge in creating this figure. Is there any way that it can be made more crisp. The combination of very small fonts and rough resolution makes it very hard to read (at least when printed).**

We have generated an improved version of Figure 6 (in section 3.1) with a crisper resolution and as large as possible font size for the labelling within it.

**Comment 5:**

**P18 L31: Need to say "northern-hemisphere winter" here (it's said in the next sentence), but it should be clear when "winter" is first used.**

We have corrected this in the manuscript. The sentence (near the beginning of Section 3.2) now reads:

'Cloud pH is constrained more by AOD in **northern-hemisphere** winter (Figure 7a) when in-cloud oxidation of $SO_2$ by ozone dominates sulphate production.'

**Comment 6:**

**P19 L13: Is nitrate not included at all in the simulations? It seems like that could throw things off a lot (which is I guess what you're saying). It would be good to make it clear that nitrate was not included (if it's not) or if it's just incorrect.**

The model does not contain a scheme for nitrate aerosol. This is stated in our description of the HadGEM3-UKCA climate model version we have used in Section 2.1. However, we agree with the reviewer that this is not clear at this point in the results section, and we have therefore revised the text to give more clarity in the point we are making. This text (in Section 3.2) now reads:

'However, it may also indicate a structural deficiency. *The low deposition rates in winter imply that PM2.5 has missing sources in winter but not in the summer, such as nitrate. Our model does not include aerosol nitrate, which (if included) would increase northern-hemisphere winter PM2.5 concentrations and weaken the constraint on dry deposition towards lower values in northern-hemisphere winter.*'

**Have the authors looked at how much the observations impact the direct aerosol \*effect\* (the radiative effect of natural and anthro aerosols relative to no aerosols) rather than the radiative forcing? I wonder how different the level of constraint would be. This isn't necessary for the paper unless the authors can do this quickly and agree that it's interesting.**

We had not looked into how our constraint using this large set of aerosol observations would impact the direct aerosol radiative effect (the radiative effect of natural and anthropogenic aerosols relative to no aerosols).

Figure 1 (below) shows the effect of the observational constraint using all measurement types on the distribution of the present-day global annual mean clear-sky aerosol radiative effect (RE; relative to no aerosol radiative effects). The distribution is constrained towards weaker (less negative) values and the 95% CI range ratio [Constrained/Unconstrained] is 0.53, which translates to a reduction in the 95% CI of 47% on constraint. This constraint is stronger than the constraints we obtain on the pre-industrial to present-day radiative forcing (RF), which we expected as the RE is more closely related to our set of measurements (all aerosol properties) than RF, for which the uncertainty depends on both aerosol and cloud-related properties (Regayre et al, 2018).

[Figure]

**Figure 1:** Effect of observational constraint using all measurement types on the probability distribution of the present-day global annual mean clear-sky aerosol radiative effect (relative to no aerosol radiative effects. The black line shows the prior (unconstrained) distribution and the red line shows the constrained distribution.

We have added the following text to the manuscript in Section 3.6 to reflect that the constraint is stronger for the direct aerosol radiative effect, however we do not include the figure as we want the focus of the paper to remain on the effect of the observational constraint on the radiative forcing.

'*Furthermore, we have found that the observational constraint on present-day (2008) global annual mean aerosol radiative effects (RE; relative to no aerosol radiative effects) is stronger than the constraint on aerosol radiative forcing. The uncertainty in the clear-sky aerosol RE is reduced by 47%. Industrial-period aerosol radiative forcing has distinct sources of uncertainty from aerosol radiative effects in the present-day atmosphere (Regayre et al., 2018), so a stronger constraint on present-day radiative effects is in line with expectations.*'

**References:**

Yoshioka, M., Regayre, L. A., Pringle, K. J., Johnson, J. S., Mann, G. W., Partridge, D. G., Sexton, D. M. H., Lister, G. M. S., Schutgens, N., Stier, P., Kipling, Z., Bellouin, N., Browse, J., Booth, B. B. B., Johnson, C. E., Johnson, B., Mollard, J. D. P., Lee, L. A. and Carslaw, K. S.: Ensembles of Global Climate Model Variants Designed for the Quantification and Constraint of Uncertainty in Aerosols and their Radiative Forcing, J. Adv. Model. Earth Syst., 11(11), 3728–3754, doi:10.1029/2019MS001628, 2019.

Regayre, L. A., Johnson, J. S., Yoshioka, M., Pringle, K. J., Sexton, D. M. H., Booth, B. B. B., Lee, L. A., Bellouin, N. and Carslaw, K. S.: Aerosol and physical atmosphere model parameters are both important sources of uncertainty in aerosol ERF, Atmos. Chem. Phys., 18(13), 9975–10006, doi:10.5194/acp-18-9975-2018, 2018.

**Response to reviewer 2 (Anonymous reviewer)**

**This manuscript investigates the constraint on aerosol forcing uncertainties by observations of AOD, PM2.5, particle number concentrations, sulphate and organic mass concentrations. The manuscript is scientifically sound, and of a high scientific quality. In my opinion, the manuscript is ready for publication in ACP, since it presents not significant shortcomings in the scientific significance or quality.**

We thank the reviewer for their support to our manuscript. As the reviewer has said our manuscript is ready for publication, no amendments have been made in response to this review.